

**Bayesian performance evaluation of evapotranspiration models for an arid region in**
**northwestern China**
Guoxiao Wei [1, 2,*], Xiaoying Zhang [3,4,*], Ming Ye [5], Ning Yue [1,2], Fei Kan [1,2]
[1] Key Laboratory of Western China's Environmental System (Ministry of Education), Lanzhou University,
China, 730000
[2] School of Earth and Environmental Sciences, Lanzhou University, China, 730000
[3] Institute of Groundwater and Earth Sciences, Jinan University, China, 510632
[4] Construct Engineering College, Jilin University, China, 130400
[5] Department of Earth, Ocean, and Atmospheric Science, Florida State University, USA, 32306
* Corresponding author: xiaoyingzh@jlu.edu.cn; gxwei@lzu.edu.cn.



**Abstract**
Evapotranspiration (ET) is a major component of the land surface process involved in energy fluxes
and balance, especially in the hydrological cycle of agricultural ecosystems. While many models have been
developed to estimate ET, there has been no agreement on which model has the best performance. In this
study, we evaluate four widely used ET models (i.e., the Shuttleworth Wallace (SW) model,
Penman-Monteith (PM) model, Priestley-Taylor and Flint-Childs (PT-FC) model, and Advection-Aridity
(AA) model) by using half-hourly ET observations obtained at a spring maize field in an arid region. The
model evaluation is based on Bayesian model comparison and ranking using the Bayesian model evidence
(BME), which balances between goodness-of-fit to data and model complexity. The BME-based model
ranking (from the best to the worst) is SW, PM, PT-FC, and AA. The residuals between observations and
corresponding model simulations are also analyzed, and the same model ranking is also obtained by using
residual-based statistics, i.e., the coefficient of determination ($R^2$), index of agreement (IA), root mean
square error (RMSE) and model efficiency (EF). The PM and SW models overestimate ET, whereas the
PT-FC and AA models underestimate ET in the study period. The four models also underestimate ET
during the periods of partial crop cover. Especially during the late maturity stage, the PT-FC and AA
models consistently produce an underestimation, and provide the worst simulated ET. As a result, at the
half-hourly time scale, the SW model is the best model and recommend as the first choice for evaluating
ET of spring maize in arid desert oasis areas.
**Keywords:** Evapotranspiration; Bayesian analysis; Penman–Monteith; Shuttleworth-Wallace; Maize
**1. Introduction**
Surface energy fluxes are an important component of Earth's global energy budget and a primary





determinant of surface climate. Evapotranspiration (ET), as a major energy flux process for energy balance,
accounts for about 60 ~ 65% of the average precipitation over the surface of the Earth. In agricultural
ecosystems, more than 90% of the total water losses are due to ET (Brutsaert, 2005). Therefore, ET
estimation is crucial to a wide range of problems in hydrology (Xu and Singh, 1998), ecology, and global
climate change (Morison et al., 2008). In practice, much of our understanding of how land surface
processes and vegetation affect weather and climate is based on numerical modeling of surface energy
fluxes and the atmospherically-coupled hydrological cycle (Bonan, 2008). Several models are commonly
used in agricultural systems to evaluate ET. The Penman-Monteith (PM) and Shuttleworth-Wallace (SW)
models are physically sound and rigorous (Zhu et al., 2013), and thus widely used to estimate ET for
seasonally varied vegetations. The models consider the relationships among net radiation, all kinds of heat
flux (such as latent heat sensible heat, and heat from soil and canopy), and surface temperature. The
Priestley-Taylor and Flint-Childs (PT-FC) model (based on radiation) and the advection-aridity (AA)
model (based on meteorological variables) have also been widely used because they require a small
amount of ground-based measurements for setting up the models (Ershadi et al., 2014). Evaluating the
performance of these four models is the focus of this study.

These ET models are generally complex for the coupling of the land surface and atmospheric

processes, and high-dimensional with a large number of parameters. Modelers are challenged by how to
compare the competing models and how to evaluate the mismatch between model simulations and
corresponding observed surface-atmosphere water flux (Legates, 1999). Moreover, how to choose a
criterion to reliably evaluate model performance is another crucial issue. Both non-Bayesian analysis
(Szilagyi and Jozsa, 2008; Vinukollu et al., 2011; Li et al., 2013; Ershadi et al., 2015) and Bayesian





analysis have been used for evaluating model performance (Zhu et al., 2014; Chen et al., 2015; Liu et al.,
2016; Zhang et al., 2017; Elshall et al., 2018; Samani et al., 2018; Zeng et al., 2018) . These quantitative
criteria used for model evaluation and selection include residual-based measures (e.g., regression line slope
and mean bias error, MBE), squared residual-based measures (e.g., coefficient of determination, $R^2$), root
mean square error (RMSE), model efficiency (EF), and index of agreement (IA). Li et al. (2013) compared
the maize ET estimates given by PM, SW and adjusted SW models under film-mulching conditions in an
arid region of China. They found that the half-hourly ET was overestimated by 17% by the SW model,
with relatively high MBE, RMSE, and lower $R^2$ and IA. In contrast, the PM and MSW models
underestimated the daily ET by 6% and 2%, respectively, during the entire experimental period of 116
days. Therefore, the performance of PM and MSW models are better than that of the SW model in their
case. Ershadi et al. (2014) evaluated the surface energy balance system (SEBS), PM, PT-JPL (modified
Priestley–Taylor model, similar to the PT-FC) and AA models. Based on the average value of EF and
RMSE, the model rank from the worst to the best was AA, PM, SEBS, and PT-JPL. Ershadi et al. (2015)
also evaluated model response to the different formulations of aerodynamic and surface resistances against
global FLUXNET data. The results showed significant variability in model performance among and within
biome types.

The Bayesian model evidence (BME), also known as marginal likelihood, measures the average fit

of a model to the data over a model's parameter space. When comparing several alternative conceptual
models, the model with the largest marginal likelihood is selected as the best model (Lartillot and Philippe,
2006). BME can thus be used for evaluating the model fit (over the parameter space) and for comparing
alternative models. In previous studies, Bayesian information criterion (BIC; Kashyap, 1982) or Kashyap



information criterion (KIC; Schwarz 1978) were used to approximate BME for reducing computational
cost of evaluating BME (Ye et al., 2004). However, these approximations have theoretical and
computational limitations (Ye et al., 2008; Xie, 2011; Schöniger et al., 2014), and a numerical evaluation
(not an approximation) of BME is necessary, especially for complex models (Lartillot and Philippe, 2006).
Lartillot and Philippe (2006) advocated the use of thermodynamic integration (TI) for estimating BME,
which is also known as path sampling (Gelman and Meng, 1998; Neal, 2000), to avoid sampling solely in
the prior or posterior parameter space. TI uses samples that are systematically generated from the prior to
the posterior parameter space by conducting path sampling with several discrete power coefficient values
(Liu et al., 2016). It is both mathematically rigorous and more accurate than the generally used harmonic
mean method (Xie et al., 2011).

While many statistical criteria have beed used to evaluate different ET models, BME has not beed

used for evaluating the ET models. It remains to be determined whether BME can be used to compare and
select the best model and whether BME can provide an unbiased view of the performance of the models.
Furthermore, most Bayesian applications have focused on the calibration of individual models and
comparison of alternative models using these statistical measures, with little attention given to the
Bayesian model comparison. Model calibration, comparison, and analysis underlying the Bayesian
paradigm has been much less used in the evaluation of ET models than in other areas of environmental
science.

In this study, the Bayesian approach was used to calibrate and evaluate the four ET models (PM, SW,

PT-FC, and AA) based on an experiment over a spring maize field in an arid area of northwest China, from
3 June to 27 September 2014. The ET models were calibrated using the DiffeRential Evolution Adaptive



Metropolis (DREAM) algorithm. The objectives of the study are as follows: (1) to compare the four
models and select the best one using BME; (2) to evaluate various general statistics such as
correlation-based measures ($R^2$), relative error measures (IA and EF), and absolute error measures (such as
RMSE and MBE) and to determine whether these methods are efficient and reasonable for evaluating the
ET models; (3) to analyze model-data mismatch for better understanding model performance. Using BME
for evaluating the ET models has not been reported in the literature. We expect that the study will not only
boost the development of model parameterization and model selection but also contribute to the
improvement of the ET models.

**2. Data and methodology**
*2.1. Description of the study area*

The experiment was conducted at Daman Superstation, located in Zhangye, Gansu, northwest China.

Daman Oasis is located in the middle Heihe River basin, which is the second largest inland river basin in
the arid region of northwest China. The midstream area of the Heihe River basin is characterized by oases
with irrigated agriculture, and is a major zone of water consumption for domestic and agricultural uses.
The annual average precipitation and temperature are 125 mm and 7.2 °C (1960–2000), respectively. The
annual accumulated temperature (>10 °C) is 3,234 °C, and the annual average potential evaporation is
about 2,290 mm. The average annual duration of sunshine is 3,106 h with 148 frost-free days. The
predominant soil type is silty-clay loam and the depth of the frozen layer is about 143 mm. The study area
is a typical irrigated agriculture region, and the major water resources are the snowmelt from the Qilian
Mountains. The maize and spring wheat are the principal crops, Maize is generally sown in late April and



harvested in mid-September and is planted with a row spacing of 40 cm and a plant spacing of 30 cm. The
plant density is about 66,000 plants per hectare.

### 2.2. Measurements and data processing

Our observation data were collected from the field observation systems of the Heihe Watershed Allied
Telemetry Experimental Research (HiWATER) project as described in Li et al (2013). The observation
period was from DOY (day of the year) 154 to DOY 270 in 2014. An open-path eddy covariance (EC)
system was installed in a maize field, with the sensors at the height of 4.5 m. Maize is the main crop in the
study region, which can supply sufficient planting area to set the EC measurements. The EC data was
logged at a frequency of 10 $H_Z$ and then processed with an average time interval of 30 min. Sensible and
latent heat fluxes were computed by the EC approach of Baldocchi (2003). Flux data measured by EC
were controlled by traditional routes, including three-dimensional rotation (Aubinet et al., 2000), WPL
(Webb-Penman-Leuning) density fluctuation correction (Webb et al., 1980), frequency response correction
(Xu et al., 2014), and spurious data removal caused by rainfall, water condensation, and system failure.
About 85% of the energy balance closure was observed in the EC data (Liu et al., 2011).
Standard hydro-meteorological variables, including rainfall, air temperature, wind speed, and wind
direction, were continuously measured at the heights of 3, 5, 10, 15, 20, 30 and 40 m above the ground.
Soil temperature and moisture were measured at heights of 2, 4, 10, 20, 40, 80, 120 and 160 cm.
Photosynthetically active radiation was measured at a height of 12 m. Net radiation, including downward
and upward and longwave radiation, was measured by a four-component net radiometer. An infrared
thermometer was installed at a height of 12 m. LAI was measured approximately every 10 days during the
growing season.

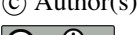


### 2.3. Model description

#### 2.3.1 Penman-Monteith (PM) model

The PM model can be formulated as following (Monteith, 1965) and most of the parameters are explained in Appendix A:

$$\lambda E = \frac{\varepsilon A + \left( \rho C_{\mathrm{p}} / \gamma \right) D_{\mathrm{a}} g_{\mathrm{a}}}{\varepsilon + 1 + g_{\mathrm{a}} / g_{\mathrm{s}}} \tag{1}$$

where $\varepsilon = \Delta / \gamma$; and $A$ is defined as:

$$A = R_{\mathrm{n}} - G \tag{2}$$

In the present study, $g_a$ is parameterized as suggested by Leuning (2008) and $g_s$ is defined as:

$$g_{\mathrm{s}} = g_{\mathrm{s}}^{\mathrm{c}} \left[ \frac{1 + \dfrac{\tau g_{\mathrm{a}}}{(\varepsilon + 1) g_{\mathrm{s}}^{\mathrm{c}}} \left[ f - \dfrac{(\varepsilon + 1)(1 - f) g_{\mathrm{s}}^{\mathrm{c}}}{g_{\mathrm{a}}} \right] + \dfrac{g_{\mathrm{a}}}{\varepsilon g_i}}{1 - \tau \left[ f - \dfrac{(\varepsilon + 1)(1 - f) g_{\mathrm{s}}^{\mathrm{c}}}{g_{\mathrm{a}}} \right] + \dfrac{g_{\mathrm{a}}}{\varepsilon g_i}} \right] \tag{3}$$

where $1-\tau$ is the fraction of the total available energy absorbed by the canopy and by the soil, and $\tau = \exp(-k_A LAI)$; $g_i$ is defined as:

$$g_i = \frac{A}{\left( \rho C_{\mathrm{p}} / \gamma \right) D_{\mathrm{a}}} \tag{4}$$

(Monteith, 1965); $g_{\mathrm{s}}^{\mathrm{c}}$ is expressed as:

$$g_{\mathrm{s}}^{\mathrm{c}} = \frac{g_{\max}}{K_{\mathrm{Q}}} In \left[ \frac{Q_{\mathrm{h}} + Q_{50}}{Q_{\mathrm{h}} \exp(-K_{\mathrm{Q}} LAI) + Q_{50}} \right] \left[ \frac{1}{1 + D_{\mathrm{a}} / D_{50}} \right] f(\theta) \tag{5}$$

where $f(\theta)$ is the factor considers water stress and is expressed as:





$$f(\theta) = \begin{cases} 1 & \theta > \theta_a \\ \dfrac{\theta - \theta_b}{\theta_a - \theta_b} & \theta_b < \theta < \theta_a \\ 0 & \theta < \theta_b \end{cases}$$ (6)
where $\theta_a$ was set as $\theta_a = 0.75\,\theta_b$. Aerodynamic conductance is calculated as:
$$g_a = \frac{k^2 u_m}{\ln\left[(z_m - d)/z_{0m}\right]\ln\left[(z_m - d)/z_{0v}\right]}$$ (7)
where the quantities $d$, $z_{0m}$ and $z_{0v}$ are calculated using $d = 2h/3$, $z_{0m} = 0.123h$ and $z_{0v} = 0.1z_{0m}$ (Allen 1998).

### 2.3.2. Shuttleworth-Wallace (SW) model

The SW model comprises a one-dimensional model of plant transpiration and a one-dimensional
model of soil evaporation. The two terms are calculated by the following equations:
$$\lambda ET = \lambda E + \lambda T = C_s ET_s + C_c ET_c$$ (8)
$$ET_s = \frac{\Delta A + \left\{\rho C_p(e_s - e_a) - \Delta r_a^s\left(A - A_s\right)\right\}/\left(r_a^a + r_a^s\right)}{\Delta + \gamma\left\{1 + r_s^s/\left(r_a^a + r_a^s\right)\right\}}$$ (9)
$$ET_c = \frac{\Delta A + \left\{\rho C_p(e_s - e_a) - \Delta r_a^c A_s\right\}/\left(r_a^a + r_a^c\right)}{\Delta + \gamma\left\{1 + r_s^c/\left(r_a^a + r_a^c\right)\right\}}$$ (10)
where the available energy input above the soil surface is defined as:
$$A_s = R_{ns} - G$$ (11)
$R_{ns}$ can be calculated by using the Beer's law relationship:
$$R_{ns} = R_n \exp(-K_A LAI)$$ (12)
The two coefficients $C_s$ and $C_c$ are obtained as follows:





$$C_s = \left\{ 1 + R_s R_a / R_c \left( R_s + R_a \right) \right\}^{-1}$$ (13)
$$C_c = \left\{ 1 + R_c R_a / R_s \left( R_c + R_a \right) \right\}^{-1}$$ (14)
where $R_c$, $R_a$, and $R_s$ are given as:
$$R_a = \left( \Delta + \gamma \right) r_a^a$$ (15)
$$R_s = \left( \Delta + \gamma \right) r_a^s + \gamma r_s^s$$ (16)
$$R_c = \left( \Delta + \gamma \right) r_a^c + \gamma r_s^c$$ (17)
Soil surface resistance is expressed as:
$$r_s^s = \exp(b_1 - b_2 \frac{\theta}{\theta_s})$$ (18)
In this study, we consider the reciprocal of bulk stomatal resistance, known as canopy conductance.
The calculation of $g_s^c$ is the same as in the PM model. The two aerodynamic resistances ($r_a^a$ and $r_a^s$) and
the boundary layer resistance (rac) are modeled following the approach proposed by Shuttleworth and
Gurney (1990).
### 2.3.3. Priestley–Taylor and Flint-Childs (PT-FC) model
The Priestley–Taylor (Priestley and Taylor, 1972) model was introduced to estimate evaporation from
an extensive wet surface under conditions of minimum advection (Stannard, 1993; Sumner and Jacobs,
2005). It is expressed as:
$$\lambda ET = \alpha_{PT} \frac{\Delta}{\Delta + \gamma} \left( R_n - G \right)$$ (19)





where $\alpha_{PT}$ is a unitless coefficient. The Priestley–Taylor model was modified by Flint and Childs (1991) to
scale the Priestley–Taylor potential ET to actual ET for nonpotential conditions (hereafter the PT-FC
model):
$$\lambda ET = \alpha \frac{\Delta}{\Delta + \gamma}\left(R_n - G\right)$$    (20)
where $\alpha$ is as a function of the environmental variables, which could be related to any process that limits
ET (e.g., soil hydraulic resistance, aerodynamic resistance, stomatal resistance); however, only soil
moisture status was considered to simplify ET estimation in the PT-FC model (Flint and Childs, 1991). In
this model, $\alpha$ is defined as:
$$\alpha = \beta_1\left[1 - \exp\left(-\beta_2 \Theta\right)\right]$$    (21)
where $\Theta$ is calculated as
$$\Theta = \frac{\theta - \theta_r}{\theta_s - \theta_r}$$    (22)
***2.3.4. Advection-aridity (AA) model***
The AA model was first proposed by Brutsaert and Stricker (1979) and further improved by Parlange
and Katul (1992). The model relies on the feedback between actual ($\lambda ET$) and potential $ET$, which assumes
that actual potential $ET$ should converge to wet surface $ET$ at wet surface conditions. Its general form is:
$$\lambda ET = \left(2\alpha_{PT} - 1\right)\frac{\Delta}{\Delta + \gamma}\left(R_n - G\right) - \frac{\gamma}{\Delta + \gamma}\frac{\rho\left(q^* - q\right)}{r_a}$$    (23)
where $\alpha_{PT}$ is the Priestley–Taylor coefficient, usually taken as 1.26 (Priestley and Taylor, 1972); and $r_a$ is
similar to that used for the Penman-Monteith model (Brutsaert and Stricker, 1979; Brutsaert, 2005; Ershadi



et al., 2014). This model is based mainly on meteorological variables and does not require any information
related to soil moisture, canopy resistance or other measures of aridity (Ershadi et al., 2014). In this study,
similar to the PT-FC model, we modified $\alpha_{PT}$ to $\alpha$, which is calculated using the same equation as in the
PT-FC model. The detailed list of symbols and physical characteristics in ET models are stated in
Appendix A.

### 2.4 BME Estimation

#### 2.4.1 Thermodynamic Integration Estimator

Estimating the BME using power posterior estimators such as thermodynamic integration (TI)
(Lartillot and Philippe, 2006) depends mainly on the likelihood $p(\mathbf{D}|\boldsymbol{\theta}, M)$ calculation. The main idea of
power posterior sampling is to define a path that links the prior to the unnormalized posterior. Thus, using
an unnormalized power posterior density
$$q_\beta(\boldsymbol{\theta}) = p(\mathbf{D}|\boldsymbol{\theta}, M)^\beta \, p(\boldsymbol{\theta}|M) \tag{24}$$
the power coefficient $\beta \in [0,1]$ is a scalar parameter for discretizing a continuous and differentiable path
linking two unnormalized power posterior densities. The unnormalized power posterior density $q_\beta(\boldsymbol{\theta})$ in
Equation (24) uses the normalizing constant $Z_\beta$ to yield the normalized power posterior density:
$$p_\beta(\boldsymbol{\theta}) = \frac{q_\beta(\boldsymbol{\theta})}{Z_\beta} \tag{25}$$
such that
$$Z_\beta = \int q_\beta(\boldsymbol{\theta}) d\boldsymbol{\theta} \tag{26}$$




The above integral takes a simplified form by the potential:
$$U(\theta) = \frac{\partial \ln q_\beta(\theta)}{\partial \beta}$$    (27)
thus, the integration can be directly estimated by following:
$$p(\mathbf{D}|M) = \frac{Z_1}{Z_0} = \exp\left\{\int_0^1 E_\theta\left[\ln p(\mathbf{D}|\boldsymbol{\theta},M)\right]d\beta\right\}$$    (28)
*2.4.2 Power posterior sampling*
The Metropolis acceptance ratio is $\alpha_k = \min\left(1,[\alpha_{k,power-postrior}\alpha_{k,prior}]\right)$ with the power posterior
ratio given by $\alpha_{k,power-posterior} = \left(\alpha_{k,posterior}\right)^{\beta_k}$. The prior probability ratio
$\alpha_{k,prior} = \Pr(\boldsymbol{\theta}_{k,new}|M)/\Pr(\boldsymbol{\theta}_{k,old}|M)$ is the ratio of the probability of the newly proposed sample
$\boldsymbol{\theta}_{k,new}$ and the probability of the previously accepted sample $\boldsymbol{\theta}_{old}$. The posterior probability ratio
$\alpha_{k,posterior} = L(\mathbf{D}|\boldsymbol{\theta}_{k,new},M)/L(\mathbf{D}|\boldsymbol{\theta}_{k,old},M)$ is the likelihood ratio of samples $\boldsymbol{\theta}_{k,new}$ and $\boldsymbol{\theta}_{k,old}$, and $\beta_k$
is the power posterior coefficient. Thus, to use the DREAM to sample any power posterior distributions
(Bayesian inference and the DREAM algorithm please see the Appendix B), the regular Metropolis
acceptance ratio $\alpha = min\left(1,[\alpha_{posterior}\alpha_{prior}]\right)$ is changed to $\alpha_k = \min\left(1,[\alpha_{k,power-postrior}\alpha_{k,prior}]\right)$ in DREAM.
Since there has been no theoretical method so far for selecting $\beta$ values (Liu et al., 2016), we
determined these values using an empirical but straightforward method. Following Xie et al. (2011), a
schedule of the power posterior coefficients $\beta_k$ is generated by
$$\beta_k = (k/K)^{1/\alpha}$$    (29)
for $k$ =0, 1, 2…, $K$. Using $\alpha = 0.3$ and $K = 20$ is a reasonable initial choice.



### 246    *2.4.3 Estimation of BME using TI*

After obtaining the power posterior samples, the corresponding likelihoods are used to estimate the
BME. This step simply requires the log-likelihoods $\log p(\mathbf{D}|\boldsymbol{\theta}_{k,i}, M)$ to be inserted into the following
equation:
$$\hat{r}_{TI} \equiv p(\mathbf{D}|M) = \exp\left(\int_0^1 y_\beta \, d\beta\right) = \exp\left(\sum_{k=0}^{K} r_{TI,k}\right) \tag{30}$$

such that
$$r_{TI,k} = (\beta_k - \beta_{k-1})\left[\frac{y_k - y_{k-1}}{2}\right] \tag{31}$$

and
$$y_k = E_\beta[\log p(\mathbf{D}|\boldsymbol{\theta}_k, M)] = \frac{1}{n}\sum_{i=1}^{n} \log p(\mathbf{D}|\boldsymbol{\theta}_{k,i}, M) \tag{32}$$

Each panel in this one-dimensional integral is given by $r_{TI,k}$ for the case of the trapezoidal rule (Eq.
31), and the summation of these panels gives the natural logarithm of BME.

### 257    *2.5 Traditional statistical evaluations*

The traditional statistics for evaluating model performance include correlation-based measures of
$R^2$ and slope, relative error measures of IA and EF, and absolute error measures of RMSE and mean bias
error (MBE) (Poblete-Echeverria and Ortega-Farias, 2009). Their definitions are as follows:
$$\text{IA} = 1 - \frac{\sum_{t=1}^{n}[O(t) - M(t)]^2}{\sum_{t=1}^{n}\left[\left|O(t) - \overline{O(t)}\right| + \left|O(t) - \overline{M(t)}\right|\right]^2} \tag{33}$$




$$EF = 1 - \frac{\sum_{t=1}^{n}[O(t) - M(t)]^2}{\sum_{t=1}^{n}\left[O(t) - \overline{O(t)}\right]^2}$$
(34)

$$RMSE = \sqrt{\frac{1}{n}\sum_{t=1}^{n}[O(t) - M(t)]^2}$$
(35)

$$MBE = \frac{1}{n}\sum_{t=1}^{n}[O(t) - M(t)]$$
(36)

where $O(t)$ is the observations and $\overline{O(t)}$ is the mean observation at time $t$, $M(t)$ is the modeled
value estimated by the posterior median parameter values, and $n$ is the total number of the observed values.

**3. Results**
*3.1 Parameter estimation*
There were five parameters $g_{max}$, $D_{50}$, $Q_{50}$, $K_Q$ and $Q_A$ in the PM model, and two additional parameters,
$b_1$ and $b_2$, in the SW model. The PT-FC and AA models include two parameters, denoted as $\beta_1$ and $\beta_2$
(Table 1). The prior probability density of each parameter was specified as a uniform distribution with the
ranges listed in Table 1. A total of 50,000 realizaitions were generated with DREAM and the calibration
period data were from DOY 154 to DOY 202. In the calculations, the chain number, $N$, was equal to the
parameter number, i.e., $N = 5$, 7, 2 and 2 for the PM, SW, PT-FC and AA models, respectively. For each
model, the first 10,000 samples were discarded as burn-in data, and the remaining 40,000 samples were
used to set up posterior density functions for each chain.
To understand the efficiency and convergence of DREAM for the ET models, Figure 1 shows the





trace plots of the G-R statistic for each of the different parameters in the PM and SW models with the
different color (PT-FC and AA models not shown). The algorithm requires about 8,000 generations to
make the G-R statistic smaller than 1.2 for the both models. Obviously, the complete mixing of the
different chains and convergence of DREAM were attained after about 620 and 450 generations for PM
and SW models, respectively. The acceptance rates for the PM and SW models were about 15.3% and
18.9%, respectively.

Histograms and cumulative distribution functions (CDFs) of the DREAM-derived marginal

distributions of the parameters are presented in Figure 2 and summarized in Table 2 by Maximum
Likelihood Estimates (MLEs), posterior medians and 95% probability intervals. The uppercase in Figures
2A-2E, 2I-2O, 2F-2G, and 2H and 2P showed histograms, and the corresponding lowercase of 2a-2e, 2i-2o,
2f-2g, and 2h and 2p show CDFs, for the PM, SW, PT-FC and AA models, respectively.

Parameter $g_{max}$ (Fig. 2A) in the PM model, parameters $g_{max}$, $K_A$, $b_1$, $b_2$ (Fig. 2I, 2M, 2N, 2O) in the SW

model, and parameter $\beta_1$ (Fig. 2F) were well constrained and occupied a relatively small range. These
parameter sample displayed a unimodal distribution, and became approximately symmetric. Parameters
$Q_{50}$, $D_{50}$, $K_Q$ and $Q_A$ (Fig. 2B-2E) in the PM model and parameters $D_{50}$, $K_Q$ in the SW model (Fig. 2K-2L)
exhibited relatively large uncertainty reductions. However, the histograms obviously deviated from
normality and tended to concentrate in the lower bounds. When the upper limits of these parameters were
decreased, similar histograms were reached (not shown) and still did not show statistically meaningful
distributions. In contrast, $Q_{50}$ was not only poorly constrained (Fig. 2J) but was also the edge-hitting
parameter in the SW model. In addition, the corresponding distributions of the same parameter in different
models were slightly different; for example, the mean of $g_{max}$ in the PM model (0.04 mm s$^{-1}$) was less than




that in the SW model (0.01 mm s$^{-1}$) (Fig. 2A and 2I; Table 2), except that $D_{50}$ in the PM and SW models
and $\beta_2$ in the PT-FC and AA models exhibited similar regions (Fig. 2C and 2K, 2G and 2P; Table 2).

### 3.2 Performance of the models

The performances of the four evaporation models were evaluated during the whole season in 2014.

The calibrated parameters of the four models were used and individual ET models were run to estimate the
half-hourly λET values. Statistical results for the performance of the models were summarized in tables as
the regression line slope, $R^2$, RMSE, MBE, IA, and EF as shown in Table 3. The regressions between
measured and modeled λET values and MBE are shown in Figures 3 and 4, respectively.

In general, the four models produced slightly better fits to the measured λET for all the seasons with

$R^2$ larger than 0.75 (Fig. 3). However, obvious discrepancies among models were detected by comparing
measured and modeled λET. According to the regression line slope and MBE, the PM model
overestimated ET by 1% with a MBE of -9.52 W m$^{-2}$, and the SW overestimates ET by 5% with a
relatively higher MBE of -19.07 W m$^{-2}$ compared to the PM model. The PT-FC and AA models tended to
underestimate λET by 9% and 8% with an MBE of 25.42 and 23.29 W m$^{-2}$, respectively. From a
comparison between the slope and MBE, the PM model performance was higher than the SW, PT-FC and
AA models, with a slope almost equal to 1 and with relatively lower MBE. The SW model was ranked
second, while the AA model was comparable to the PT-FC, but slightly higher, and was ranked third.
However, if $R^2$, RMSE, IA, and EF were used to evaluate the model performances, the SW model had the
best overall performance with $R^2$=0.83, RMSE= 76.34 W m$^{-2}$, IA = 0.95 and EF = 0.79. The second-best
model was the PM model with $R^2$ = 0.76, RMSE = 85.38 W m$^{-2}$, IA = 0.93 and EF = 0.74.The PT-FC
performance was ranked third with $R^2$ = 0.75, RMSE = 94.39 W m$^{-2}$, IA = 0.92 and EF = 0.68, while the



AA model ranked fourth with R$^2$ = 0.75, RMSE = 95.09 W m$^{-2}$, IA = 0.92 and EF = 0.67. Based on the
analysis of these traditional statistical criteria, the performances of the PT-FC and AA models yielded
similar results. The observed and modeled λET for the four ET models were tightly grouped along the
regression lines (Figure 3), and the PT-FC and AA models had similar modeled ET values with a similar
degree of point scattering along the regression lines (Figure 3c-3d).

Figure 4 shows that large seasonal variations were exist in MBE for the four ET models. From the

variations of the MBE, the estimated λET values for all models were generally lower than the measured
values before the early jointing stage of maize growth (DOY 154-177, left dashed line) and after the late
maturity stage (DOY 256-265, right dash line) with the corresponding LAI < 2.5 m$^2$ m$^{-2}$. More positive
MBE values for the PT-FC and AA models after the late maturity stage indicated their underestimated
performances; however, these estimations appeared even more consistent with a symmetrical scattering of
points along the 0-0 line (Figure 4c, 4d) during DOY 177-256 with LAI > 2.5 m$^2$ m$^{-2}$.
*3.3 Comparison of the models using BME*

Since there was no theoretical method so far for selecting *β* values, we determined these values using

empirical but straightforward methods. For any different power coefficient of $\beta \in [0,1]$, a sample was
drawn from the distribution $p_\beta$ (Eq. 25) through running DREAM. Figure 5 showed the evolution of *ln*
$p(D|\theta, M)$ for the four models as a function of *β* for a dataset covering the entire period. The potential
values of the PM model increased from -6533.02 (the logarithm of the prior likelihood) to -6290.71, and
the potential values increased from -6544.49 to -6016.17 for the SW model. In addition, the potential
values increased from -6708.02 to -6361.76 for the PT-FC model and from -7732.98 to -7033.32 for the
AA model. Table 3 showed that the estimated BME is -6300.5 natural log units (nits) for the PM model,





-6025.1 nits for the SW model, -6366.8 nits for the PT-FC model, and -7042.8 nits for the AA model. The
BME for the SW model was substantially larger than that for the other three models, and the BME for the
AA model was the smallest. Although the parameters of the PM model were less than for the SW model,
the potential evolution of the SW model was substantially different to that of the PM model. In summary,
the PT-FC and AA models, consisting of the same number of parameters, had similar potential patterns of
evolution with the coefficient $\beta_k$. Although adding more $\beta_k$ values may improve the BME estimation, it was
not undertaken because of the computational cost. For each $\beta_k$ value, 150,000 DREAM simulations were
large enough to ensure convergence.
**4. Discussion**
*4.1 Parameter uncertainty analysis*
With regard to the efficiency of the DREAM algorithm, the acceptance rates of the PM (15.33%) and
SW (18.94%) models were much higher than some MCMC algorithms which used in the previous studies,
like 0.019% in the population Monte Carlo sampling algorithm (Sadegh et al., 2014). This was a large
improvement in search efficiency, which in large part resulted from its ability to sample groups of variable
in turn. Furthermore, this method ran multiple chains in parallel and adaptively updated the scale and
orientation of the proposal distribution (Vrugt et al., 2008). Therefore, the DREAM scheme substantially
improved not only the convergence, but also its sampling efficiency for ET models.
The results showed that the DREAM algorithm successfully reduced the assumed prior uncertainties
from the large number of parameters in the four models. The well-constrained parameters were those that
had significant contribution. For example, the ecophysiological parameter $g_{max}$, in both the PM and SW



models, the maximum stomatal conductance of leaves, and the soil surface resistance parameters $b_1$ and $b_2$
in the SW models, all had large influences on the evaluated ET. Thus, their effects were relatively
independent compared to the other meteorological parameters in the models. The posterior mean value of
$g_{max}$ (0.04 m s$^{-1}$) in the PM model from our study was close to that (0.05 m s$^{-1}$) reported in northwestern
China (Li et al., 2013; Zhu et al., 2014), but $g_{max}$ (0.01 m s$^{-1}$) in the SW model was less than the reported
value. The estimated posterior means for $b_1$ and $b_2$ were different ($b_1$ = 9.3, $b_2$ = 6.2) from those for maize
suggested by Zhu et al. (2014) using the same equation of soil surface resistance ($r_s^s$). Though Zhu et al.
(2014) concluded that the responses of $g_s^c$ to VPD and LAI calculated using the modified Leuning model
were close to those using Javis model (Jarvis, 1976), Li et al. (2015) showed that the performance of PM
model was different using the two canopy resistance formula. Therefore, the different results of parameters
$b_1$ and $b_2$ between our study and the previous study by Zhu et al. (2014) were mainly due to the usage
ofdifferent canopy conductance models.
For edge-hitting parameters, their uncertainties may be the outcome of model biases or EC-measured
ET data, or the characteristic time scale of parameters govern processes that was not exactly on the order
of   half-hours (Braswell et al., 2005). For example, $Q_{50}$ and $D_{50}$ govern changes in visible radiation flux
and humidity deficit at which stomatal conductance at its half maximum value, which may change over a
shorter or longer time scale rather than half-hours. $K_Q$ was another parameter that cannot be well
constrained, and this may be resulted from either the estimated ET was insensitive to these parameters, or
there were correlations between the parameters. We expected a complementary correlative relationship
between the visible radiation flux and extinction coefficient for shortwave radiation, which indicated that
the information in EC-measured ET data was insufficient to separate these parameters, and therefore the



parameters cannot be constrained separately.

The sensitive parameters (such as $g_{max}$, $b_1$ and $b_2$) were just corresponding to the well-constrained

parameters. Therefore, the major parameters in PM and SW models were well optimized, except that
several parameters ($Q_{50}$ and $K_Q$) appeared to be not well constrained. In addition, the posterior parameter
bounds exhibited a larger reduction using the DREAM algorithm compared with other studies using the
Metropolis–Hasting algorithm (Zhu et al., 2014). This further demonstrated that DREAM can efficiently
handle problems involving high-dimensionality, multimodality, nonlinearity, and local optima.

In general, parameters related to soil surface resistance in the SW model were well evaluated, while

parameters related to canopy surface resistance in PM and SW models were poorly estimated. Therefore,
using a reliable canopy surface resistance equation in the ET model was crucial for improving its
performance. In addition, in our study, the traditional approach was used to quantify the uncertainty which
assumed that the uncertainty mainly came from the parameter uncertainty. However, this method did not
explicitly consider errors in the input data and model structural inadequacies. This was unrealistic for real
applications, and it was desirable to develop a more reliable inference method to treat all sources of
uncertainty separately and appropriately (Vrugt et al., 2008). Moreover, simultaneous direct measurement
by micro-lysimeter of sap flow and daily soil evaporation will further help to constrain the model
parameters.
*4.2 Evaluation and selection of the models*

In this study, the traditional statistical measures and BME were chosen to evaluate and compare the

performance of four ET models. From the respective composition of these measures, the statistical




measures can be divided into residual-based measures (such as regression slope and MBE) and
squared-residual-based measures (such as $R^2$, RMSE, IA, and EF). Table 3 shows the values evaluated by
BME method, residual-based and squared-residual-based measures. By comparison, the estimates obtained
within the same measure (residual-based or squared-residual-based) were congruent. For example, slope
and MBE have similar results in the residual-based measures. However, the results from different kind of
measures were incongruent; for example, PM model outperformed SW model according to the
residual-based measures, but PM model was worse than SW model based on the squared-residual-based
measures. The comparative analysis showed a consistency between BME and the squared-residual-based
statistics, whereas residual-based criteria were obvious disagreement with the BME measures. It revealed
that the more complex SW model was the best model based on the BME and squared-residual-based
statistics. The rank order of overall performance of the models from best to worst was: SW, PM, PT-FC,
and AA model.

Previous studies had shown that BME evaluated by the TI provided estimates similar to the true

values and selected the true model if the true model was included within the candidate models (Marshall et
al., 2005; Lartillot and Philippe, 2006). Meanwhile, some argured that Bayesian analysis would choose the
simplest model (Jefferys and Berger, 1992; Xie et al., 2011) because of the best trade-off between good fit
with data and model complexity (Schöniger et al., 2014). In this case, the most complex SW model had the
highest BME and was chosen as the best-behaved model. This likely resulted from the fact that the
complex SW model was indeed the most reliable model among the alternative ET models. SW model was
a two-layer model, and estimated soil evaporation and plant transpiration separately, but PM model was a
single-layer model while the plant transpiration and soil evaporation cannot be separated (Monteith, 1965).





The PT-FC model was a simplified model of PM, and it only required meteorological and radiation
information (Priestley and Taylor, 1972), whereas AA model only relied on the feedback between actual
ET and potential ET (Brutsaert and Stricker, 1979). Based on these physical mechanisms and processes for
these ET models, the rank order of the models was reasonable.

The estimates showed that the maximum values of $R^2$, IA and EF, and the minimum value of RMSE,

all selected the most complex SW model as the best performing model. The results indicated that the SW
model was the best performing model evaluated by squared-residual-based measures, which resulted from
the ability of the model to fit the measured data, irrespective of model complexity. It was interesting to
note that both the squared-residual-based measures and the BME consistently yielded the same rank order.
Although the squared-residual-based measures seemed to identify a reasonable rank order, this had often
not been the case, since the simple traditional statistical measures were known to usually provide a biased
view of the efficacy of a model (Kessler and Neas, 1994; Legates and McCabe, 1999). In addition,
sensitivity to outliers was associated with these measures and leads to relatively high values due to the
squaring of the residual terms (Willmott, 1981). Furthermore, these traditional statistical measures ignored
the priors, without penalizing model complexity, which was in fact used in a Bayesian analysis. The
dimensionality (model's parameter space) not only affected model evaluation by BME (Schöniger et al.,
2014) but it may also affect the evaluation using traditional statistical measures. Here, two-dimensional
models of PT-FC and AA provided identical estimates of $R^2$ and IA. This was most likely because both the
PT-FC and AA models had the same dimensions and a similar model structure, whereas BME estimates
remain well-behaved for the two ET models. Marshall et al. (2005) argued that EF would provide an
incorrect conclusion, and Samani et al. (2018) suggested that RMSE also selected the complex model as

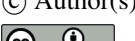


the best performing model. Thus, we deduced that SRB measures are also problematic. As for slope and
MBE, these residual-based measures were obvious disagreement with the BME measure. Part of the lower
values of slope and MBE may be counter balanced by the higher values of slope and MBE, thus these
criterias provided an erroneous and unreliable model evaluation. Therefore, the squared-residual-based and
residual-based measures were not certain to provide reasonable results in terms of model ranking.
*4.3 Analysis of model-data mismatch*

Conceptual and structural inadequacies of the hydrological model and measurement errors of the

model input (forcing) and output (calibration) data introduced errors in the estimated parameters and model
simulations (Laloy, 2014). Hydrological systems were indeed heavily input driven and errors in forcing
data can dramatically impair the quality of calibration results and model output (Bardossy and Das, 2008;
Giudice, 2015). Measurement errors were raised for a variety of reasons, including unreasonable
gap-filling in rainy days; dew and fog; inadequate areal coverage of point-scale soil water measurement;
mechanical limitations of the EC system; and inaccurate measurements of wind-speed, soil water, radiation
and vapor pressure deficit. ET process was described using equations that can only capture parts of the
complex natural processes and the model structures were an inherent simplification of the real system.
These inadequacies can thus lead to biased parameters and implausible predictions.

In our study, the results indicated that the PM and SW models overestimated the half-hourly ET

compared to the measured ET. Several studies also indicated that the ET values were overestimated by the
PM model (Fisher et al., 2005; Ortega-Farias et al., 2006; Li et al., 2015) and the SW model (Li et al., 2013;
Li et al., 2015; Zhang et al., 2008). Possible reasons for the inaccurate estimates included the following: (1)
Anisotropic turbulence with weak vertical and strong horizontal fluctuation leads to energy imbalance. The



total turbulent heat flux was lower by ~10–30% compared to the available energy in many land surface
experiments (Tsvang et al., 1991; Beyrich et al., 2002; Oncley et al., 2007; Foken et al., 2010) and influx
networks (Franssen et al., 2010). Liang et al. (2017) also showed an energy imbalance result in the
semiarid area in China, and indicated that the energy balance closure ratio ranged from 0.52 to 0.90 during
the daytime, whereas it was about 0.25 during night-time. However, the measured ET only included
vertical flux and not horizontal flux, leading to the measured ET being lower than that of modeled ET by
the PM and SW models using the available energy. (2) The absence of a mechanistic representation of the
physiological response to plant hydrodynamics cause it difficult for the available ET models to resolve the
dynamics of intradaily hysteresis, producing patterns of diurnal error, while the imbalance or lack of
between-leaf water demand and soil water supply imposes hydrodynamic limitations on stomatal
conductance (Thomsen et al., 2013; Zhang et al., 2014; Matheny et al., 2014). Li et al. (2015) also
concluded that neglecting the restrictive effect of the soil on water transport in empirical canopy resistance
equations can result in large errors in the partial canopy stage. However, these equations can simulate ET
accurately under the full canopy stage (Alves and Pereira, 2000; Katerji and Rana, 2006; Katerji et al.,
2011; Rana et al., 2011). Li et al. (2015) showed the PM model combined with the canopy resistance
overestimated maize ET during the partial and dense canopy stages by 16% and 13%, respectively
(Leuning, 2008). Moreover, the PM model coupled with the canopy resistance overestimated vineyard ET
during the entire growth stage by 29% (Leuning, 2008).

The estimated ET for the PT-FC and AA models was generally lower than the measured values during

the entire season. In addition, the four models also underestimated the ET during periods of partial cover
(LAI < 2.5 m$^2$ m$^{-2}$). Especially during the late maturity stage, the PT-FC and AA models consistently



underestimated ET and provided the worst simulated ET. The underestimation probably resulted from the
following: (1) Non classical situations, such as the oasis effect, may occur in the study area. Strong
evaporation from the moist ground and plants results in latent heat cooling. However, this upward latent
heat flux was opposed by a downward sensible heat flux from the warm air to the cool ground, and thus the
latent heat flux was positive while the sensible heat flux is negative. Therefore, the latent heat flux can be
greater in magnitude than the solar heating, because of the additional energy extracted from the warm air
by evaporation (Stull, 1988). (2) Lack of mechanistic representation of rainfall interception in ET models
probably also led to inaccurate simulation on shortly after rainy days. Bohn and Vivoni (2016) found that
evaporation of canopy interception accounted for 8% of the annual ET across the North American
monsoon region.

Comparing the AA and PT-FC models, the former included forcing data of available radiation, soil

water content and relative humidity, but the PT-FC model only requires available radiation and soil water
content and was independent of relative humidity. However, the similar statistical results and similar
degrees of MBE scatter indicated that relative humidity has little influence on the AA model simulation.
The consistent and consecutive underestimation of ET by the PT-FC and AA models during the late
maturity stage showed that the model-data disagreement is caused mainly by regional advection and
rainfall interception, because atmospheric processes and thermally-induced circulation can only occur at
certain times and during certain days. Therefore, we suggested that the consistent underestimation of ET
by the PT-FC and AA models primarily results from conceptual and structural inadequacies, energy
imbalance, and soil water stress. Although the PM and SW models shared a common theoretical basis and
the PT-FC model was the simplification of the PM model, these models performed significantly differently.



Part of the overestimation of ET by the PM and SW models, caused by coupling with the canopy resistance,
may be offset by underestimation caused by energy imbalance and soil water stress. However,
underestimation of ET by the PT-FC and AA models cannot be counterbalanced by overestimation during
the later maturity stage because the PT-FC and AA models are independent of the canopy resistance.
Consequently, the half-hourly patterns of errors in the prediction of ET by the PM and SW models were
characterized by symmetry and a low degree of scatter, but the PT-FC and AA models exhibited consistent
and asymmetrical error patterns.

By contrast, other studies showed that the PM model (Kato et al., 2004) and the SW model (Chen et

al., 2015) underestimated half-hourly ET. As for the PT-FC and AA models, while some studies reported
that the PT-JPL (Zhang et al., 2017) and AA model showed an overall poor performance, however, other
studies have indicated that the AA method performed well for both maize and canola crops (Liu et al.,
2012). Therefore, the performance of the four ET models appears to vary not only for different crops and
locations (Zhu et al., 2014) but also for different meteorological, physiological and soil conditions.
Moreover, the performance was also related to the stage of crop growth.

**5. Conclusions**

This study illustrated the use of the Bayesian approach for the statistical analysis and model selection

of four widely used ET models. BME can be used to rank the alternative models in our study, although
numerical evaluation of BME is computationally expensive particularly for high-dimensional models.
Bayesian model comparison identified the SW model as the best ET model. Although the
squared-residual-based measures, including $R^2$, IA, RMSE, and EF, provide a congruent model ranking




with that of BME, it must be noted that these squared-residual-based measures do not allow using prior
information for comparing the models. We advocated that caution is needed when using these statistical
methods, and that BME should be used instead. In contrast, residual-based measures disagree with the
BME measure, and thus these measures can not be used for evaluating model performance.

The model–data mismatches were analyzed to facilitate model improvement after using Bayesian

model calibration and comparison. The results indicated that model–data mismatches are mainly resulted
from energy imbalance caused by anisotropic turbulence, the absence of a mechanistic representation of
the physiological response to plant hydrodynamics, and additional energy induced by advection processes.
Among them, energy imbalances and additional energy were related to forcing data error rather than to an
unreasonable model structure. Thus, understanding the process of the physiological response to plant
hydrodynamics, such as developing or selecting more reasonable and process-based canopy resistance
models, was essential for improving the performance of evapotranspiration models. Overall, in our study,
the applications of Bayesian calibration, Bayesian model evaluation and analysis of model–data
mismatches, provided a promising framework for reducing uncertainty and improving the performance of
ET models.

**Author contribution**

Guoxiao Wei and Xiaoying Zhang designed the experiments. Ning Yue and Fei Kan carried them out.

Ming Ye developed the model selection scheme. Guoxiao Wei performed the simulations. Guoxiao Wei
and Xiaoying Zhang prepared the manuscript with contributions from all co-authors.

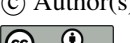




## Competing interests

The authors declare that they have no conflict of interest.

## Acknowledgments

We thank Ying Guo, Huihui Dang, Jun Dong for the data collection and analysis. This work was
funded by the National Natural Science Foundation of China (Nos. 41471023). The third author was
supported in part by DOE Early Career Award DE-SC0008272 and National Science Foundation-Division
of Earth Science Grant 1552329. All observed data used in this study are from Heihe Watershed Allied
Telemetry Experimental Research (HiWATER). We thank all the staff who participated in HiWATER field
campaigns. Considerate and helpful comments by anonymous reviewers have considerably improved the
manuscript.

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

**Appendix A: List of symbols and physical characteristics in ET models**

| Symbol | Description |
|---|---|
| $A$ | Available energy for the whole canopy ($Wm^{-2}$) |
| $A_s$ | Available energy (W $m^{-2}$) |
| $R_n$ | Net radiation fluxes into the canopy (W $m^{-2}$) |
| $R_{ns}$ | Net radiation flux into the substrate (W $m^{-2}$) |
| $G$ | Soil heat flux (W $m^{-2}$) |
| $\lambda ET$ | Sum of the latent heat flux from the crop ($\lambda T$) and soil ($\lambda E$) (W $m^{-2}$) |
| $ET_c$ | Canopy transpiration (W $m^{-2}$) |
| $ET_s$ | Soil evaporation (W $m^{-2}$) |
| $C_c$ | Canopy resistance coefficient (dimensionless) |
| $C_s$ | Soil surface resistance coefficient (dimensionless) |
| $k_A$ | Extinction coefficient for available energy |
| $LAI$ | Leaf area index |
| $Q_{50}$ | Visible radiation flux (W $m^{-2}$) |
| $D_{50}$ | Vapor pressure deficit (kPa) |
| $D_a$ | Vapor pressure deficit at the reference height ($D_a = e_s - e_a$) (kPa) |
| $Q_h$ | Flux density of visible radiation at the top of the canopy    (W $m^{-2}$) |
| $K_Q$ | Extinction coefficient |
| $K_A$ | Extinction coefficient |
| $f$ | Fraction of evaporation soil and total evaporation |
| $\lambda$ | Latent heat of water evaporation (MJ $kg^{-1}$) |
| $\Delta$ | Slope of the saturated vapour pressure curve (Pa $K^{-1}$) |
| $\gamma$ | Psychrometric constant (kPa $K^{-1}$) |
| $\rho$ | Density of air (kg $m^{-3}$) |
| $k$ | Karman constant (0.41) |
| $e_s$ | Saturated vapor pressure (kPa) |
| $e_a$ | Actual vapor pressure (kPa) |
| $q^*$ | Saturation-specific humidity at air temperatur (kg $kg^{-1}$) |
| $q$ | Specific humidity of the atmosphere (kg $kg^{-1}$) |
| $b_1$ | Empirical constant    (s $m^{-1}$) |
| $b_2$ | Empirical constant    (s $m^{-1}$) |
| $\beta_1$ | empirical constant |
| $\beta_2$ | empirical constant |
| $\theta$ | Soil water content ($m^3$ $m^{-3}$) |
| $\theta_a$ | Critical water content at which plant stress starts ($m^3$ $m^{-3}$) |
| $\theta_b$ | Water content at the wilting point ($m^3$ $m^{-3}$) |





| | |
|---|---|
| $\theta_r$ | Residual soil water content (m$^3$ m$^{-3}$) |
| $\theta_s$ | Saturated water content (m$^3$ m$^{-3}$) |
| $\Theta$ | Relative water saturation |
| $d$ | Zero plane displacement height (m) |
| $z_m$ | Height of the wind speed and humidity measurements (3 m) |
| $z_{0m}$ | Roughness length governing the transfer of momentum (m) |
| $z_{0v}$ | Roughness length governing the transfer of water vapor (m) |
| $h$ | Canopy height (m) |
| $u_z$ | Wind speed at height $z_m$ (m s$^{-1}$) |
| $g_a$ | Aerodynamic conductance (m s$^{-1}$) |
| $g_s$ | Surface conductance (m s$^{-1}$) |
| $g_{max}$ | Maximum stomatal conductance of leaves at the top of the canopy (m s$^{-1}$) |
| $g_s^c$ | Canopy conductance (m s$^{-1}$) |
| $r_a$ | Aerodynamic resistance (s m$^{-1}$) |
| $r_a^a$ | Aerodynamic resistance between canopy source height and a reference level (s m$^{-1}$) |
| $r_a^s$ | Aerodynamic resistance between the substrate and the canopy source height (s m$^{-1}$) |
| $r_a^c$ | Bulk boundary layer resistance of the vegetation element in the canopy (s m$^{-1}$) |
| $r_s^s$ | Surface resistance of the canopy (s m$^{-1}$); |
| $r_s^c$ | Bulk stomatal resistance of the canopy (s m$^{-1}$) |


**Appendix B: Bayesian inference and the DREAM algorithm**
The posterior probability distribution of the parameter is calculated by Bayes' theorem:
$$\pi(\theta\,|\,D,M) = \frac{\pi(\theta\,|\,M)\,p(D|\theta,M)}{p(D|M)} \qquad \text{(A1)}$$
where $\pi(\theta\,|\,M)$ represents the prior density of $\theta$ under model $M$; $p(D|\theta,M)$ is the joint likelihood of
model $M$ and its parameters $\theta$; and
$$p(D\,|\,M) = \int p(D\,|\,\theta,M)\,p(\theta\,|\,M)\,d\theta \qquad \text{(A2)}$$
is the marginal likelihood, or Bayesian model evidence (BME).
The likelihood function, $p(D|\theta, M)$, used for parameter estimation, is specified according to the
distributions of observation errors. Error $e(t)$ in each observation $D(t)$ at time $t$ is expressed by





$$e(t) = D(t) - f(t) \qquad (A3)$$


. Assuming $e(t)$ follows a Gaussian distribution with a zero mean, and the likelihood function can be
expressed as

$$p(D|\theta) = \prod_{t=1}^{n} \frac{1}{\sqrt{2\pi}\sigma} e^{-\frac{[e(t)]^2}{2\sigma^2}} \qquad (A4)$$


where $n$ is the number of observations and $\sigma$ represents the error variances.

In this study, we used the DREAM algorithm (Vrugt et al., 2008, 2009) to explore the ET models'

parameter space and to estimate BME. The DREAM sampling scheme is an adaptation of the global
optimization algorithm of a shuffled complex evolution metropolis (SCEM-UA). This algorithm was
described in more detail in Vrugt et al. (2008, 2009).

**List of Tables**
**Table 1** Prior distributions and parameter limits for the PM, SW, PT-FC and AA models. The values are
derived from the literature.
**Table 2** Maximum Likelihood Estimates (MLEs), Mean Estimates, 95% High-Probability Intervals
(Lower Limit, Upper Limit).
**Table 3 S**lope and coefficient of determination ($R^2$) of regression between measured and modeled
half-hourly evapotranspiration values, and statistics of root mean square error (RMSE), mean bias error
(MBE), index of agreement (IA), model efficiency (EF) and Logarithm of BME for the four ET models.





**Table 1** Prior distributions and parameter limits for the PM, SW, PT-FC and AA models. The values are
derived from the literature.

| Parameter | Description | Prior range PM | Prior for SW | Prior for PT and AA | References |
|---|---|---|---|---|---|





| | | Lower | upper | Lower | upper | Lower | upper | |
|---|---|---|---|---|---|---|---|---|
| $g_{max}$ (mm s$^{-1}$) | maximum stomatal conductance | 0 | 50 | 0 | 50 | | | Kelliher et al. (1995) |
| $Q_{50}$ (W m$^{-2}$) | visible radiation flux | 10 | 50 | 10 | 50 | | | Leuning et al. (2008) |
| $D_{50}$ (kPa) | vapor pressure deficit | 0.5 | 3 | 0.5 | 3 | | | Leuning et al. (2008) |
| $K_Q$ | extinction coefficient | 0 | 1 | 0 | 1 | | | Leuning et al. (2008) |
| $K_A$ | extinction coefficient | 0 | 1 | 0 | 1 | | | Leuning et al. (2008) |
| $b_1$ (s m$^{-1}$) | empirical constant | | | 4.5 | 11.3 | | | Sellers et al. (1992) |
| $b_2$ (s m$^{-1}$) | empirical constant | | | 0 | 8 | | | Sellers et al. (1992) |
| $\beta_1$ | empirical constant | | | | | 0.5 | 1.5 | Flint et al. (1991); |
| $B_2$ | empirical constant | | | | | 0.1 | 10 | Barton. (1979) |




**Table 2** Maximum Likelihood Estimates (MLEs), Mean Estimates, 95% High-Probability Intervals
(Lower Limit, Upper Limit).

| Parameter | Posterior for PM | | | Posterior for SW | | | Posterior for PT and AA | | |
|---|---|---|---|---|---|---|---|---|---|
| | MLE | Mean | CI | MLE | Mean | CI | MLE | Mean | CI |
| $g_{max}$ (mm s$^{-1}$) | 0.04 | 0.04 | (0.03, 0.04) | 0.01 | 0.01 | (0.005, 0.012) | | | |
| $Q_{50}$ (W m$^{-2}$) | 49.96 | 48.52 | (39.73, 49.74) | 47.49 | 40.32 | (11.02, 48.99) | | | |
| $D_{50}$ (kPa) | 3.00 | 2.87 | (1.92, 2.97) | 2.98 | 2.88 | (2.26, 2.98) | | | |
| $K_Q$ | 1.00 | 0.99 | (0.911, 0.998) | 0.99 | 0.88 | (0.06, 0.98) | | | |
| $K_A$ | 1.00 | 0.98 | (0.822, 0.995) | 0.12 | 0.12 | (0.074, 0.184) | | | |
| $b_1$ (s m$^{-1}$) | | | | 4.51 | 4.57 | (4.52, 4.96) | | | |
| $b_2$ (s m$^{-1}$) | | | | 0.39 | 0.57 | (0.07, 1.38) | | | |





| | | | |
|---|---|---|---|
| $\beta_1$ | $1.1^a$ | $1.098^a$ | $(1.06, 1.16)^a$ |
| | $1.5^b$ | $1.499^b$ | $(1.492, 1.499)^b$ |
| $\beta_2$ | $10.00^a$ | $9.75^a$ | $(7.97, 9.95)^a$ |
| | $10.00^b$ | $9.94^b$ | $(9.44, 9.99)^b$ |

[a] PT-FC model; [b] AA model.



**Table 3** Slope and coefficient of determination ($R^2$) of regression between measured and modeled
half-hourly evapotranspiration values, and statistics of root mean square error (RMSE), mean bias error
(MBE), index of agreement (IA), model efficiency (EF) and Logarithm of BME for the four ET models.

| Model | Slope | $R^2$ | RMSE | MBE | IA | EF | BME |
|---|---|---|---|---|---|---|---|
| PM | 1.01 | 0.76 | 85.38 | -9.52 | 0.93 | 0.74 | -6300.5 |
| SW | 1.05 | 0.82 | 76.34 | -19.07 | 0.95 | 0.79 | -6025.1 |
| PT-FC | 0.91 | 0.75 | 94.39 | 25.42 | 0.92 | 0.68 | -6366.8 |
| AA | 0.92 | 0.75 | 95.09 | 23.29 | 0.92 | 0.67 | -6390.3 |








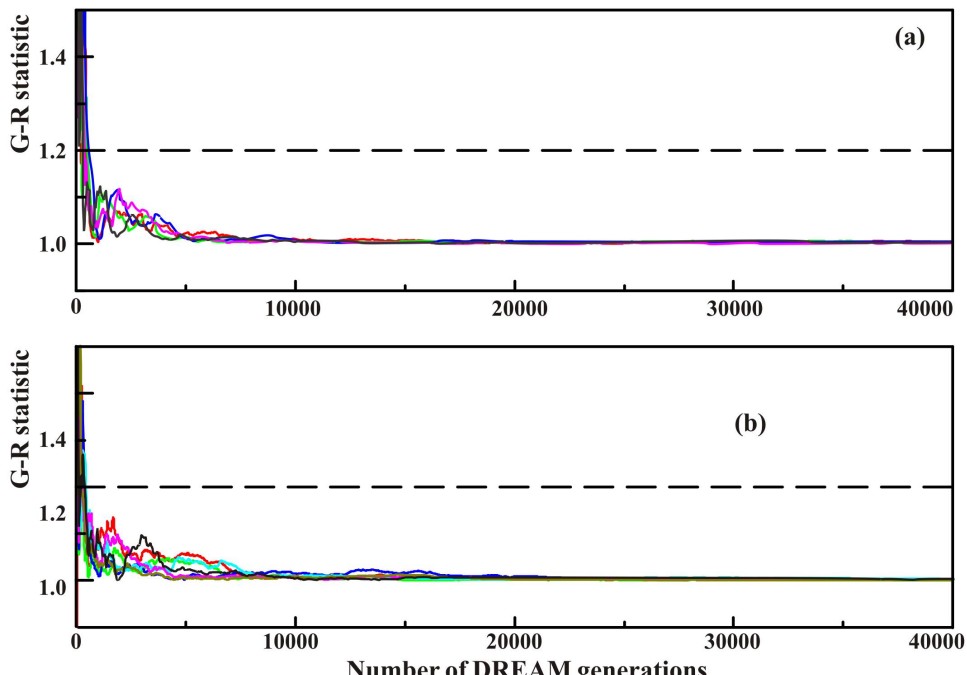


**Figure 1** Trace plots of the G-R statistic of Gelman and Rubin (Gelman and Rubin, 1992) using DREAM

for the PM model (a) and (b) the SW model. Different parameters are coded with different colors. The

dashed line denotes the default threshold used to diagnose convergence to a limiting distribution.







**Figure 2** Uppercase of 2A-2E, 2I-2O, 2F-2G, and 2H and 2P show histograms, and corresponding

lowercases of 2a-2e, 2i-2o, 2f-2g, and 2h and 2p show CDFs for the PM, SW, PT-FC and AA models,

respectively. Thee histograms and the CDFs are constructed from the one chain and 40000 generations

simulated using DREAM. The $y$ axes represent the prespecified limits of the parameters.





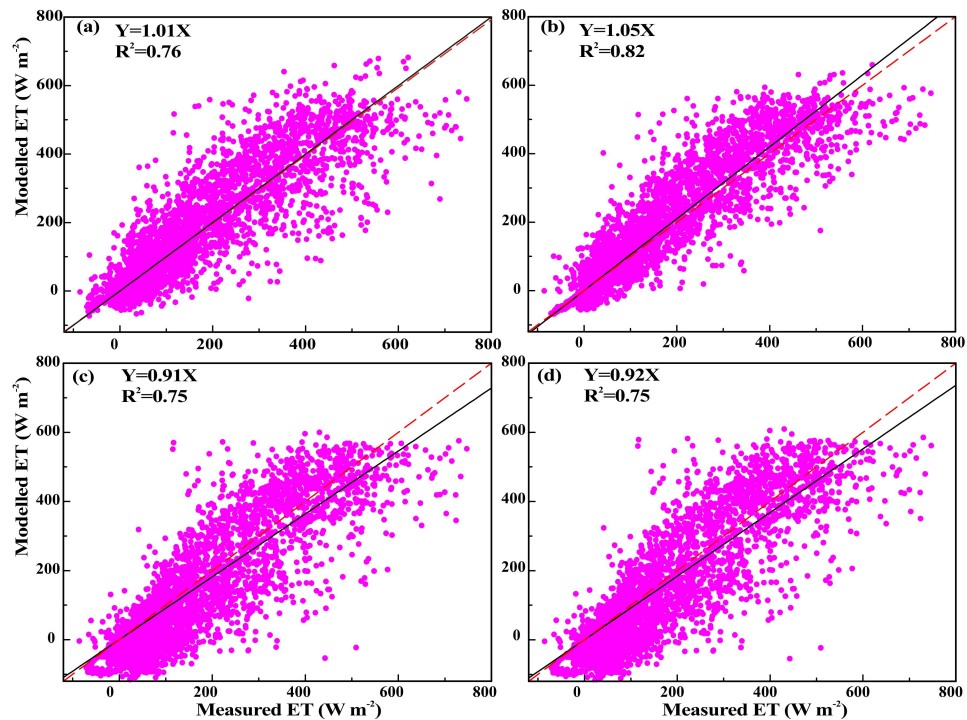


**Figure 3.** Regression between measured and modeled half-hourly evapotranspiration values produced by
different models: (a) PM, (b) SW, (c) PT-FC and (d) AA. The regressions are: Y = 0.99X ($R^2$ = 0.76), Y =
1.05X ($R^2$ = 0. 82), Y = 0.91X ($R^2$ = 0.75), and Y = 0.92X ($R^2$ = 0.75) for the PM, SW, PT-FC and AA
models, respectively.





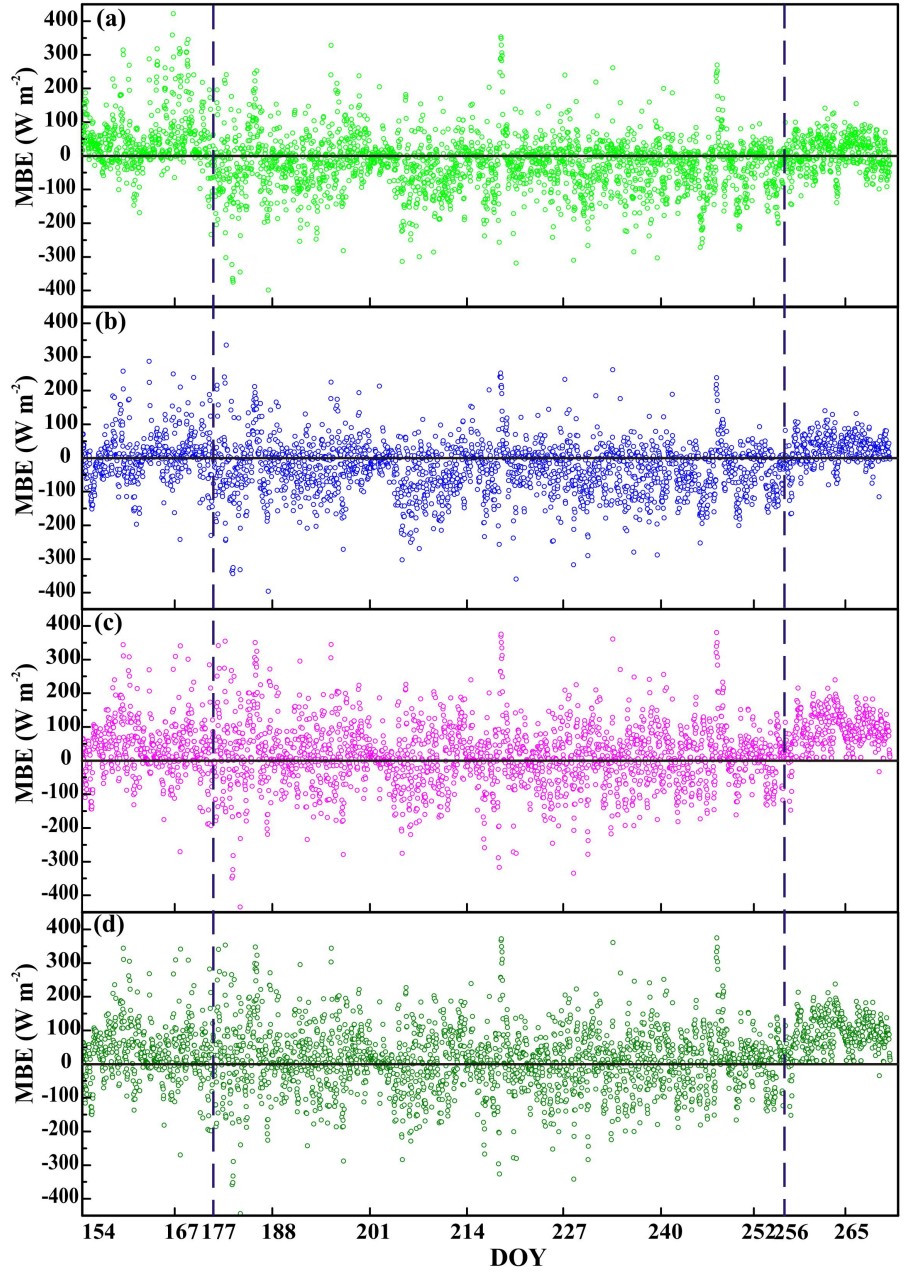


**Fig**

**ure 4.** Mean bias error (MBE) of predicted and observed ET (W m⁻²) values for (a) PM, (b) SW, (c) PT-FC
and (d) AA models from DOY 154 to DOY 270. Parameters used for prediction are estimated by DREAM



with the dataset for the calibration period from DOY 154 to DOY 202.

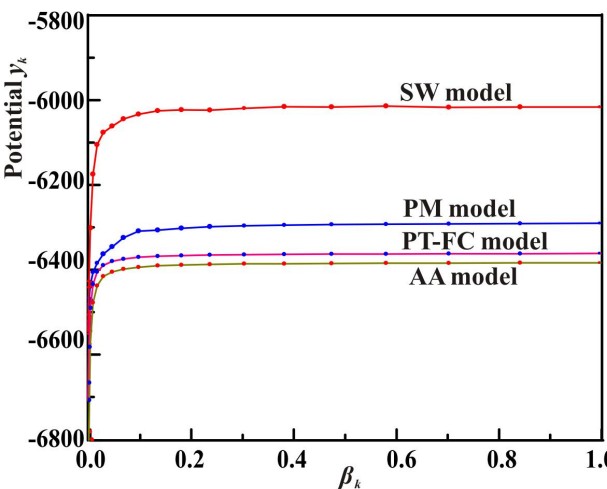


**Figure 5.** Variation of the mean posterior expectation of the potential $y_k$ (equation (36)) with $\beta_k$ (power
coefficient in equation (33)) for the PM, SW, PT-FC and AA models.