# Peer review of "Bayesian performance evaluation of evapotranspiration models based on eddy"

_Hydrology and Earth System Sciences, 2018_

## Referee Comment (RC1) · Anonymous Referee #1 · 16 Nov 2018

In this manuscript, the authors did a lot of work in using the Bayesian techniques to analyze ET models. This work could be interesting to the hydrological community. General comments: 1. Some efforts are needed to emphasize the significance of the work. For the objective (1), what is the purpose of selecting the best model using BME, to improve model prediction? If the purpose is to improve model prediction, did the authors try Bayesian model averaging? Based on the results, some models are underestimate, some models are overestimate, it is possible that model averaging could give a better prediction performance. For the objective (2), theoretically we know these statistics only measure model fit without considering model complexity, so they are not as robust as BME. And we know these statistics can be efficiently calculated, so

there is no need for testing. Please justify the objective (2). I think, objective (3) is very meaningful, I would like to see more analysis on the model-data mismatch to improve model development and model performance. 2. In several places of the manuscript, the logic is not very clear. The English writing needs improvement.

Specific comments: 1. Abstract, I think including some insights obtained from the numerical experiments in the abstract would attract more audience and make this work more meaningful. 2. Line 35-36, the SW model performs best in this study area, but may not be the best in other areas. For example, in Li et al., (2013)'s study, PM performed better than SW in estimation of maize. Please justify the statement that SW should be the first choice for evaluating ET of spring maize in arid desert oasis areas. 3. Line 93-94, BME can be used to compare and select the best-performing model. This is well-known and not a hypothesis that needs to be determined. What do the authors mean by saying "an unbiased view"? 4. Line 95-97, the first part of the sentence says Bayesian applications have focused on comparison of alternative models, but the second part of the sentence says that little attention has been given to the Bayesian model comparison. The sentence is self-contradictory. Please clarify. 5. Line 277, for each chain? I thought you total have 40,000 samples from all chains. In addition, Line 848, from one chain? Please clarify. 6. Line 280-282, based on Figure 1, DREAM needs far less than 8000 generations to make the GR statistic smaller than 1.2. Also, based on Figure 1's x-axis scale, it is hard to tell "obviously" the chain converged after about 620 and 450 generations. 7. Figure 1. In Figure 1(b) the position of the dash line is not at 1.2. The position of the label (b) is not aligned well with the label (a). 8. Figure 2. If the authors cannot get more information from the CDFs than the histograms, I suggest deleting the CDFs which make Figure 2 busy and confusing. Also, I would like to see more discussion about Figure 2; what insights the authors can obtain from these plots? 9. Line294-297, I found the discussion of the figure 2 is confusing. I think, the figure 2 says the histograms tend to concentrate in the upper bounds, not the lower bounds. Also, the authors should increase the upper limits of these parameters not decrease, because the histograms are concentrated in the upper bounds. 11. Line

355-356, what do the author mean by saying "to sample groups of variable in turn"?

Technical corrections: 1. Line 29, obstained –> obtained 2. Line 92, beed –> been

---

## Referee Comment (RC2) · Anonymous Referee #2 · 25 Mar 2019

This paper studies an important problem of ET estimations: which model has the best performance to predict ET. I believe this work provides useful insights to improve our understanding of ET model selection. I would be in favor of publication after the authors addressed the comments given below. Comments: 1. The grammar of this paper needs some improvements, some grammar errors and ambiguous sentences can be found. 2. The universality of this study and its conclusions need to be clarified since the study area and methodology are both very spatial and temporal specific. PS, are the conclusions valid under other conditions or not? 3. Following the last comment, is it possible to provide results for other study areas or using other time scales? This will provide strong evidences to support the conclusions. 4. I am not sure I can agree with

some conclusions, for example, the one in lines 531-532, the authors suggest prioritizing BME over other measurements, but BME can also provide inaccurate results.

---

## Referee Comment (RC3) · Anonymous Referee #3 · 27 Mar 2019

Dear Editors and Authors,

Sorry, as I don't have enough time to carefully go through the work, specific comments are only focused on the Introduction part of the paper.

**General Comments**

This work encloses a lot of contents and seems interesting to assess the ET model having the best performances in terms of the criteria the authors select. Language issues should be fully checked throughout the entire text before publication in HESS. Novelty of the paper should be better emphasized rather than "BME has not been used for evaluating the ET models". Model complexity for each model should be better described. For example, authors can directly introduce number of parameters with uncertainties in their experiment? According to my comments, a minor revision is suggested.

**Specific comments**

1. The Abstract is out of organization. It seems to me that you never mention the model complexity but always write "underestimation" or "overestimate" to explain why the SW is the best one.

2. Lines 25-27: unclear, please rephase this sentence

3. It is unclear for me why 'SW' is best one from the abstract.

4. Line37: please check the symbols

5. Simulate ET or estimate ET? Please be very sure of this word.

6. Line 41: add a reference

7. Lines 55-56: unclear, please rephase this sentence.

8. Lines 62-63: 'These quantitative criteria' refer to what?

9. Line 70: performances

10. Line 71: remove 'the' from 'the SW model'

11. Lines 71-72: please rephase this sentence

12. Line 73: should be model ranking? Please check the terminology.

13. Lines 75-76: unclear, significant variability of model performances?

14. Lines 92-93: been?

15. Lines 102-103: add a reference

---

## Author Comment (AC1) · 19 Apr 2019

Responds to the comments: Referee #1: General comments: 1. Response to comment: For the objective (1), what is the purpose of selecting the best model using BME, to improve model prediction? If the purpose is to improve model prediction, did the authors try Bayesian model averaging? Based on the results, some models are underestimate, some models are overestimate, it is possible that model averaging could give a better prediction performance. Response: We believe this comment is very important for considerably improve our manuscript. Our original idea was to identify which model is optimal for ET prediction, and then to improve the model with the Bayesian

model averaging. However, our result showed that model SW obtained a weight of 100%. This means that BME assigned a weight of 100% to the competing model and the weights of 0% to other three alternative models, and thus, the Bayesian model averaging prediction is also the SW prediction. Other studies on hydrological model selection have yielded similar results in that one model obtained an weight of close to 100% (e.g., Meyer et al., 2007; Lu et al., 2013; Schöniger et al., 2007). Therefore, Bayesian model averaging was not used in our study.

2. Response to comment: For the objective (2), theoretically we know these statistics only measure model fit without considering model complexity, so they are not as robust as BME. And we know these statistics can be efficiently calculated, so there is no need for testing. Please justify the objective (2). Response: Yes, this comment is quite useful. We could change the original objective (1) and (2) to "(1) to calibrate ET model parameters using the diffeRential evolution adaptive metropolis (DREAM) algorithm (Vrugt et al., 2009); (2) to identify which parameters had a greater impact on the model performance and to explain why the selected optimal model performed best".

3. Response to comment: objective (3) is very meaningful. I would like to see more analysis on the model-data mismatch to improve model development and model performance. Response: Considering the Reviewer's suggestion, we will add more discussions in 4.3 "Analysis of model-data mismatch".

4. Response to comment: In several places of the manuscript, the logic is not very clear. The English writing needs improvement. Response: It is really true as you suggested that our manuscript needs the revision of English sentence. After the revision according to the reviewer's comments, the manuscript will be edited by the professional editing services.

Specific comments: 1. Response to comment: Abstract, I think including some insights obtained from the numerical experiments in the abstract would attract more audience and make this work more meaningful. Response: Considering the Reviewer's suggestion, we added and revised the content in abstract. The ET model parameters were calibrated using the entitled differential evolution adaptive Metropolis (DREAM) algorithm; the optimal model was selected using Bayesian model evidence (BME), which was implemented using the mathematically rigorous thermodynamic integration method. The estimated parameters were analyzed to identify which parameters had a greater impact on the models' performance and explain why the optimal model performed best. The discrepancies between observations and model estimates were evaluated using traditional error metrics in order to analyze the shortcomings of the different ET models and find ways to improve their performance. The results indicate that DREAM can effectively infer the model parameters. Although the SW model with seven parameters is more complex than the other three models, the BME criterion still selects SW as the best model. This is because the structure of the SW model is more physically rigorous, its parameters have greater impact and their sensitivity is well constrained. Moreover, the results show that energy imbalance and energy interaction between canopy and surface can bias the model estimates significantly.

2. Response to comment: 2. Line 35-36, the SW model performs best in this study area, but may not be the best in other areas. For example, in Li et al., (2013)'s study, PM performed better than SW in estimation of maize. Please justify the statement that SW should be the first choice for evaluating ET of spring maize in arid desert oasis areas. Response: We changed this statement to "We anticipate that our study could provide a basis for improving ET models".

3. Response to comment: Line 93-94, BME can be used to compare and select the best-performing model. This is well-known and not a hypothesis that needs to be determined. What do the authors mean by saying "an unbiased view"? Response: We originally considered there are several competed methods can select the best-performing model. But as the reviewer said, this is "well-known", thus we decided to delete this sentence.

4. Response to comment: Line 95-97, the first part of the sentence says Bayesian applications have focused on comparison of alternative models, but the second part of the sentence says that little attention has been given to the Bayesian model comparison. The sentence is self-contradictory. Please clarify. Response: Our original intention is to say that Bayesian applications have focused calibration of individual models but the model comparison is still conducted using traditional statistical criteria. We are sorry that our expression was not very clear. We will modify the words and make it clearer.

5. Response to comment: Line 277, for each chain? I thought you total have 40,000 samples from all chains. In addition, Line 848, from one chain? Please clarify. Response: Yes, this referred the 40,000 samples from one chain. Total have 40,000 multiplied by N (chain number) samples from all chains.

6. Response to comment: Line 280-282, based on Figure 1, DREAM needs far less than 8000 generations to make the GR statistic smaller than 1.2. Also, based on Figure 1's x-axis scale, it is hard to tell "obviously" the chain converged after about 620 and 450 generations. Response: Thanks for the comments. We decided to change the sentence "DREAM needs far less than 8000 generations to make the GR statistic smaller than 1.2 for the both models " to "The algorithm needs about 8,000 generations to make the G-R statistic close to 1.0 for the both models." In addition, we will delete the statement "Obviously, the complete mixing of the different chains and convergence of DREAM were attained after about 620 and 450 generations for PM and SW models, respectively".

7. Response to comment: Figure 1. In Figure 1(b) the position of the dash line is not at 1.2. The position of the label (b) is not aligned well with the label (a). Response: Thanks for the comment. We will redraw the Figure 1. This can see in the modified Figure 1 .

8. Response to comment: Figure 2. If the authors cannot get more information from the CDFs than the histograms, I suggest deleting the CDFs which make Figure 2 busy and confusing. Also, I would like to see more discussion about Figure 2; what insights the

authors can obtain from these plots? Response: Thanks for the comment. We have redraw the Figure 2 as in the attached file. The main insights summarized as following: gs and modeled ET in PM model are relatively insensitive to Q50, D50 and Kq. Hence, these parameters could not be well constrained. The calculation of in SW model is the same as in PM model, and thus, and modeled ET for SW model are also insensitive to parameters of Q50, D50, Kq. Therefore, these three parameters were also not be well constrained in SW model. In addition, for edge-hitting parameters, their uncertainties may be also the outcome of model biases or EC-measured ET data errors. Although the ecophysiological parameter gmax is a variable in the equation in both PM and SW models, but this parameter is sensitive to and has large influences on the evaluated ET. Its effect is relatively independent compared to the other meteorological parameters in the models, and therefore this parameter was well specified in SW model. The parameter Ka is insensitive to gs and modeled ET. In contrast, Ka is contained in equation of net radiation flux into the substrate (Eq.12) in SW model. From the above analysis, we could see that Ka not only involved the distribution of energy between the canopy and the soil surface but also the energy imbalance. Therefore, parameter Ka has a great influence on the performance of the SW model.

9. Response to comment: Line294-297, I found the discussion of the figure 2 is confusing. I think, the figure 2 says the histograms tend to concentrate in the upper bounds, not the lower bounds. Also, the authors should increase the upper limits of these parameters not decrease, because the histograms are concentrated in the upper bounds. Response: Thanks for the comment. We will revise this sentence.

10. Response to comment: Line 355-356, what do the author mean by saying "to sample groups of variable in turn"? Response: The sentence should be "to sample one or groups of variable in turn". We will revise this sentence and delete all the confusing words.

11. Technical corrections: Line 29, obstained –> obtained. Line 92, beed –> been Response: We will correct these type errors. Once again, thank you very much for

your comments and suggestions.

Please also note the supplement to this comment:
https://www.hydrol-earth-syst-sci-discuss.net/hess-2018-430/hess-2018-430-AC1-supplement.pdf

─────────────────────────

[Figure]

Fig. 1.

[Figure]

[Figure]

Fig. 2.

---

## Author Comment (AC2) · 19 Apr 2019

Referee #2: Comments: 1. Response to comment: The grammar of this paper needs some improvements, some grammar errors and ambiguous sentences can be found. Response: It is really true as you suggested that our manuscript needs the revision of English sentence. After the revision according to the reviewer's comments, the manuscript will be edited by the professional translation services.

2. Response to comment: The universality of this study and its conclusions need to be clarified since the study area and methodology are both very spatial and temporal specific. PS, are the conclusions valid under other conditions or not? Response: The

ET models and BME model selection can be applied to other conditions as long as the required data can be obtained. Although there are many studies on ET model evaluation, their conclusions about model ranking are all based on traditional error metrics. Just as you said, the conclusion about whether SW model is optimal selected by BME method under other conditions still needs further confirmation. We will add relevant contents in this section.

3. Response to comment: Following the last comment, is it possible to provide results for other study areas or using other time scales? This will provide strong evidences to support the conclusions. Response: It is meaningful to provide results for other study areas or using other time scales to support the conclusions. We've been looking for reliable data from other study area or from other crops for BME model selection to confirm whether the SW model is the optimal model under other conditions. We are trying our best to connect with some agriculture institutes. However, it is difficult to obtain the required data by ET models, especially the soil water contents. So far, we have requite a data-set but the data quality has not been validated yet. And thus, we will perform a deeper research on different crops when the data is available in the future study.

4. Response to comment: I am not sure I can agree with some conclusions, for example, the one in lines 531-532, the authors suggest prioritizing BME over other measurements, but BME can also provide inaccurate results. Response: This is true but not contradict. The BME is the best measurement in our test. However, as a method, itself is not perfect especially when applied in the practical field. To clear express our opinion, the confusing words will be modified in the following process if possible. Once again, thank you very much for your comments and suggestions.
* * *

---

## Author Comment (AC3) · 19 Apr 2019

General Comments: 1. Response to comment: Language issues should be fully checked throughout the entire text before publication in HESS. Response: Thanks for the comment, our manuscript will be edited by the specialist of native speaker.

2. Response to comment: Novelty of the paper should be better emphasized rather than "BME has not been used for evaluating the ET models". Response: Thanks for the comments. We will change the original statement to "Most applications of Bayesian methods have focused on the calibration of individual models, while the comparison of alternative models continues to be performed using traditional error metrics. More

generally, Bayesian approaches to model calibration, comparison, and analysis have been used far less in the evaluation of ET models than in other areas of environmental science.".

3. Response to comment: Model complexity for each model should be better described. For example, authors can directly introduce number of parameters with uncertainties in their experiment? Response: The SW model with seven parameters is more complex than the PM model with five parameters, and the PT-FC and AA models each with two parameters, which is consistent with the commonly accepted view that including additional parameters increases complexity and improves the model performance. Considering the comments, we will add the explanations in this section. Specific comments: 1. Response to comment: The Abstract is out of organization. It seems to me that you never mention the model complexity but always write "underestimation" or "overestimate" to explain why the SW is the best one. Response: Thanks for the comments. Detailed comparison will be added in the abstract to keep it within the required range. Although the SW model with seven parameters is more complex than the other three models, the BME criterion still selects SW as the best model. A revised abstract will be given for further review.

2. Response to comment: Lines 25-27: unclear, please rephase this sentence. Response: Considering the comments, we will consider to change the original sentence to "The ET model parameters were calibrated using the entitled differential evolution adaptive Metropolis (DREAM) algorithm; the optimal model was selected using Bayesian model evidence (BME), which was implemented using the mathematically rigorous thermodynamic integration method.".

3. Response to comment: It is unclear for me why 'SW' is best one from the abstract. Response: Considering the comments, we added the sentence to explain why SW is the best one in abstract. Although the SW model with seven parameters is more complex than the other three models, the BME criterion still selects SW as the best model. This is because the structure of the SW model is more physically rigorous, its

parameters have greater impact and their sensitivity is well constrained.

4. Response to comment: Line37: please check the symbol. Response: We will check the corresponding symbols.

5. Response to comment: Simulate ET or estimate ET? Please be very sure of this word. Response: Thanks for the comment, and we will change "simulate ET" to "estimate ET".

6. Response to comment: Line 41: add a reference. Response: Thanks for the comments, the reference from "Brutsaert, 2005" will be added in the paper.

7. Response to comment: Lines 55-56: unclear, please rephase this sentence. Response: We decided to rephase this sentence as "These ET models are generally complex because of the coupling of the land surface and atmospheric processes and high-dimensional".

8. Response to comment: Lines 62-63: 'These quantitative criteria' refer to what? Response: These quantitative criteria refer to the " residual-based metrics (such as regression slope and MBE) and squared-residual-based metrics (such as R2, RMSE, IA, and EF)". This part will be revised to clearer illustrate our discussion.

9. Response to comment: Line 70: performances. Response: We will check it in the manuscript.

10. Response to comment: Line 71: remove 'the' from 'the SW model' Response: This will be removed from the corresponding sentence.

11. Response to comment: Lines 71-72: please rephase this sentence. Response: The sentence "Ershadi et al. (2014) evaluated the surface energy balance system (SEBS), PM, PT-JPL (a modified Priestley-Taylor model), and AA models" will be better.

12. Response to comment: Line 73: should be model ranking? Please check the terminology. Response: Ye, this should be "model ranking".

13. Response to comment: Lines 75-76: unclear, significant variability of model performances? Response: Thanks for the comments, we will modify this part in the following process.

14. Response to comment: Lines 92-93: been? Response: We will check the words in the corresponding line.

15. Response to comment: Lines 102-103: add a reference Response: The reference "(Vrugt et al., 2009)" will be added. Once again, thank you very much for your comments and suggestions.
* * *

---

## Author Response (AR1)

**Manuscript Number:** hess-2018-430

**Title**: Bayesian performance evaluation of evapotranspiration models:a case study based on eddy covariance system of a maize field in northwestern China

**Corresponding Author:** Xiaoying Zhang

**Authors:** GuoxiaoWei, Xiaoying Zhang, Ming Ye , Ning Yue, Fei Kan

**Dear Editor,**

On behalf of my co-authors, we thank you very much for giving us an opportunity to revise our manuscript. We are grateful to the editors and reviewers for their positive and constructive comments and suggestions on our manuscript (hess-2018-430) entitled "Bayesian performance evaluation of evapotranspiration models for an arid region in northwestern China".

We have studied reviewer's comments carefully, and revised the manuscript thoroughly to address the comments. The revision is marked in red in the revised manuscript. We have tried our best to revise our manuscript according to the comments. Attached please find the revised version, which we would like to submit for your kind consideration.

We would like to express our great appreciation to you and reviewers for comments on our paper. Looking forward to hearing from you.

Thank you and best regards.

Yours sincerely,

Xiaoying Zhang

Corresponding author:

Name Xiaoying Zhang

E-mail: xiaoyingzh@jlu.edu.cn

**List of Responses**

**Dear Editors and Reviewers:**

Thank you for your letter and for the reviewers' comments concerning our manuscript entitled "Bayesian performance evaluation of evapotranspiration models for an arid region in northwestern China" (hess-2018-430). These comments are all valuable and very helpful for revising and improving our paper, as well as the important guiding significance to our researches. We have studied comments carefully and have made corrections which we hope meet with approval. Revised portion are marked in red in the paper. The main corrections in the paper and the responds to the reviewer's comments are followed.

Thank you and best regards.

Yours sincerely,

Xiaoying Zhang

Corresponding author:

Name: Xiaoying Zhang

E-mail: xiaoyingzh@jlu.edu.cn

**Responds to the comments:**

**Referee #1:**

**General comments:**

**1. Comment**: *For the objective (1), what is the purpose of selecting the best model using BME, to improve model prediction? If the purpose is to improve model prediction, did the authors try Bayesian model averaging? Based on the results, some models are underestimate, some models are overestimate, it is possible that model averaging could give a better prediction performance.*

**Response**: We believe this comment is very important for considerably improving our manuscript. Our original idea was to identify which model is optimal for ET prediction, and then to improve the model with the Bayesian model averaging. However, our result showed that model SW obtained a weight of 100%. This means that BME assigned a weight of 100% to the competing model and the weights of 0% to other three alternative models, and thus, the Bayesian model averaging prediction is also the SW prediction. Other studies on hydrological model selection have yielded similar results in that one model obtained an weight of close to 100% (e.g., Meyer et al., 2007; Lu et al., 2013; Schöniger et al., 2007). Therefore, Bayesian model averaging was not used in our study.

**2. Comment**: *For the objective (2), theoretically we know these statistics only measure model fit without considering model complexity, so they are not as robust as BME. And we know these statistics can be efficiently calculated, so there is no need for testing. Please justify the objective (2).*

**Response**: We have changed the original objective (1) and (2) to "(1) to calibrate ET model parameters using the diffeRential evolution adaptive metropolis (DREAM) algorithm; (2) to identify which parameters had a greater impact on the model performance and to explain why the selected optimal model performed best". These changes can be seen at P5, L120-121.

**3. Comment**: *objective (3) is very meaningful. I would like to see more analysis on the model-data mismatch to improve model development and model performance.*

**Response**: Considering the Reviewer's suggestion, we added some sentences in 4.1 "Parameter uncertainty analysis", which can be seen at P18-P19, L439-453, and in 4.3 "Analysis of model-data mismatch", which can be seen at P24, L597-601.

**4. Comment**: *In several places of the manuscript, the logic is not very clear. The English writing needs improvement.*

**Response**: It is true as you suggested that our manuscript needs the revision of English sentence. After the revision according to the reviewer's comments, the manuscript have been edited by the professional translation services.

**Specific comments:**

**1. Comment**: *Abstract, I think including some insights obtained from the numerical experiments in the abstract would attract more audience and make this work more meaningful.*

    **Response**: Considering the Reviewer's suggestion, we added and revised the content in abstract. These changes can be seen at P1, L19-23.

**2. Comment**: *2. Line 35-36, the SW model performs best in this study area, but may not be the best in other areas. For example, in Li et al., (2013)'s study, PM performed better than SW in estimation of maize. Please justify the statement that SW should be the first choice for evaluating ET of spring maize in arid desert oasis areas.*

    **Response**: We have changed this statement to "The mismatch analysis indicated that explicit treatment of energy imbalance and energy interaction will be the primary way to further improve ET model performance." The change can be seen at P2, L38-39.

**3. Comment**: *Line 93-94, BME can be used to compare and select the best-performing model. This is well-known and not a hypothesis that needs to be determined. What do the authors mean by saying "an unbiased view"?*

    **Response**: We have deleted this sentence at P4, L111.

**4. Comment**: *Line 95-97, the first part of the sentence says Bayesian applications have focused on comparison of alternative models, but the second part of the sentence says that little attention has been given to the Bayesian model comparison. The sentence is self-contradictory. Please clarify.*

    **Response**: Our original intention is to say that Bayesian applications have focused calibration of individual models but the model comparison is still conducted using traditional statistical criteria. Our expression was not very clear. We changed this statement as "Most applications of Bayesian methods have focused on the calibration of individual models, while the comparison of alternative models continues to be performed using traditional error metrics." This changes can be seen at P4, L107-108.

**5. Comment**: *Line 277, for each chain? I thought you total have 40,000 samples from all chains. In addition, Line 848, from one chain? Please clarify.*

    **Response:** This referred the 40,000 samples from one chain. Total have 40,000 multiplied by *N* (chain number) samples from all chains. Please see P13, L298-299; P32, L874; P35, L905.

**6. Comment**: *Line 280-282, based on Figure 1, DREAM needs far less than 8000 generations to make the GR statistic smaller than 1.2. Also, based on Figure 1's x-axis scale, it is hard to tell "obviously" the chain converged after about 620 and 450 generations.*

    **Response:** Thanks for the comment. We changed the sentence "DREAM needs far less than 8000 generations to make the GR statistic smaller than 1.2 for the both models " to "The algorithm needs about 8,000 generations to make the G-R statistic close to 1.0 for the both models." In addition, we deleted statement "Obviously, the complete mixing of the different chains and convergence of DREAM were attained after about 620 and 450 generations for PM and SW models, respectively". The change can be seen at P13-14, L302-304.

**7. Comment**: *Figure 1. In Figure 1(b) the position of the dash line is not at 1.2. The position of the label (b) is not aligned well with the label (a).*

    **Response:** Thanks for the comment. We have redrawn the Figure 1. This can be seen Figure 1.

**8. Comment**: Figure 2. If the authors cannot get more information from the CDFs than the histograms, I suggest deleting the CDFs which make Figure 2 busy and confusing. Also, I would like to see more discussion about Figure 2; what insights the authors can obtain from these plots?

    **Response:** We have redrawn the Figure 2. This can be seen Figure 2.

    The main insights summarized as following: $g_s$ and modeled ET in PM model are relatively insensitive to $Q_{50}$, $D_{50}$ and $K_q$. Hence, these parameters could not be well constrained. The calculation of $g_s^c$ in SW model is the same as in PM model, and thus, $g_s^c$ and modeled ET for SW model are also insensitive to parameters of $Q_{50}$, $D_{50}$, $K_q$. Therefore, these three parameters were also not be well constrained in SW model. In addition, for edge-hitting parameters, their uncertainties may be also the outcome of model biases or EC-measured ET data errors. Although the ecophysiological parameter $g_{max}$ is a variable in the $g_s^c$ equation in both PM and SW models, but this parameter is sensitive to $g_s^c$ and has large influences on the evaluated ET. Its effect is relatively independent compared to the other meteorological parameters in the models, and therefore this parameter was well specified in SW model. The parameter $K_a$ is insensitive to $g_s$ and modeled ET. In contrast, $K_a$ is contained in equation of net radiation flux into the substrate (Eq.12) in SW model. From the above analysis, we could see that $K_a$ not only involved the distribution of energy between the canopy and the soil surface but also the energy imbalance. Therefore, parameter $K_a$ has a great influence on the performance of the SW model.

    The revisions and changes can be seen P14, L314-321; P14, L326-330 and P17, L397-412.

**9. Comment**: *Line294-297, I found the discussion of the figure 2 is confusing. I think, the figure 2 says the histograms tend to concentrate in the upper bounds, not the lower bounds. Also, the authors should increase the upper limits of these parameters not decrease, because the histograms are concentrated in the upper bounds.*

    **Response:** Thanks for the point. We have revised this sentence. The change can be seen at P14, L315-321.

**11. Comment**: *Line 355-356, what do the author mean by saying "to sample groups of variable in turn"?*

    **Response:** The sentence should be "to sample one or groups of variable in turn". We have removed some sentence and this can be seen P17, L388-393.

**12. Technical corrections**: *Line 29, obstained –> obtained. Line 92, beed –> been*

    **Response:** We have corrected the words already.

    **Once again, thank you very much for your comments and suggestions.**

**Referee #2:**

**Comments:**

**1. Comment**: *The grammar of this paper needs some improvements, some grammar errors and ambiguous sentences can be found.*

    **Response**: It is really true as you suggested that our manuscript needs a language improvement. After the revision according to the reviewer's comments, the manuscript have been edited by the professional translation services.

**2. Comment**: *The universality of this study and its conclusions need to be clarified since the study area and methodology are both very spatial and temporal specific. PS, are the conclusions valid under other conditions or not?*

    **Response**: The ET models and BME model selection can be applied to other conditions as long as the required data can be obtained. Although there are many studies on ET model evaluation, their conclusions about model ranking are all based on traditional error metrics. Just as you said, the conclusion about whether SW model is optimal selected by BME method under other conditions still needs further confirmation. We have added relevant contents at P24, L594-596.

**3. Comment**: *Following the last comment, is it possible to provide results for other study areas or using other time scales? This will provide strong evidences to support the conclusions.*

    **Response:** It is really true that providing results for other study areas or using other time scales would be very useful for providing strong evidences to support the conclusions. We've been looking for reliable data from other study area or from other crops for BME model selection to confirm whether the SW model is the optimal model under other conditions. However, it is difficult to obtain the required data by ET models, especially the soil water contents. So far, we haven't got the requited data yet. And thus, Thanks for the comment that we are not able to provide results for other study areas or using other time scales for BME model selection by now.

**4. Comment**: *I am not sure I can agree with some conclusions, for example, the one in lines 531-532, the authors suggest prioritizing BME over other measurements, but BME can also provide inaccurate results.*

    **Response:** We think this is true, and we deleted statement "and that BME should be used instead", and reorganized the original sentences. Please see P25, L609-615.

    **Once again, thank you very much for your comments and suggestions.**

**Referee #3:**

**General Comments:**

**1. Comment**: *Language issues should be fully checked throughout the entire text before publication in HESS.*

    **Response**: It is really true as you suggested that our manuscript needs the revision of English sentence. After the revision according to the reviewer's comments, the manuscript have been edited by the professional translation services.

**2. Comment**: *Novelty of the paper should be better emphasized rather than "BME has not been used for evaluating the ET models".*

    **Response**: Thanks for the comment. We have changed the original statement to "Currently, ET model selection and comparison have been still conducted using traditional error metrics. It is known that error metrics are not adequate to provide reasonable result of model ranking for disregarding model complexity (Marshall et al., 2005; Samani et al., 2018). The focus of this study is to use a Byesian approach to evaluate the performance of the PM, SW, PT-FC, and AA models, which is a novelty contribution of this study." These changes can be seen at P3, L81-85.

**3. Comment**: *Model complexity for each model should be better described. For example, authors can directly introduce number of parameters with uncertainties in their experiment?*

    **Response:** Considering the comments, we have described the number of parameters of each model at P13, L289-291 and added a sentences "The results illustrate that with the addition of parameters, the model complexity and the model performance are both increased." at P16, L382-383.

**Specific comments:**

**1. Comment**: *The Abstract is out of organization. It seems to me that you never mention the model complexity but always write "underestimation" or "overestimate" to explain why the SW is the best one.*

    **Response:** We have reorganized the Abstract. These can be seen P1-P2, L19-39.We also have added sentence "Although the SW model with seven parameters is sophisticated, it's good fitting to observations can counterbalance its higher complexity." These can be seen P2, L35-36.

**2. Comment**: *Lines 25-27: unclear, please rephase this sentence.*

    **Response:** Considering the comments, we have reorganized the abstract again, and changed the original sentence to "The parameters in each model were first calibrated using DiffeRential Evolution Adaptive Metropolis (DREAM) algorithm, and then were analyzed to identify their impacts on the model performance. The Bayesian model evidence (BME) approach, was further adopted to select the optimal model by incorporating the mathematically rigorous thermodynamic integration algorithm." These modifications can be seen at P1, L19-23.

**3. Comment**: *It is unclear for me why 'SW' is best one from the abstract.*

   **Response:** Considering the comments, we have added the sentences "Our results revealed that the extinction coefficient was the most significant parameter in the ET models. It was not merely partitioning the total available energy into the canopy and surface, but also including the energy imbalance correction. The extinction coefficient is well constrained in the SW model and poorly constrained in the PM model, but not considered in PT-FC and AA models" to explain why SW is the best one in abstract. Please see P2, L31-35.

**4. Comment**: *Line37: please check the symbol.*

   **Response:** We have checked the symbol at P2, L40.

**5. Comment**: Simulate ET or estimate ET? Please be very sure of this word.

   **Response:** We have changed some "estimate ET" to "simulate ET".

**6. Comment**: *Line 41: add a reference.*

   **Response:** Thanks for the comments. We have added corresponding reference "(Brutsaert, 2005)". Please see at P2, L45.

**7. Comment**: *Lines 55-56: unclear, please rephase this sentence.*

   **Response:** We rephased this sentence as "These ET models are generally complex, because of for the coupling of the land surface and atmospheric processes, and high-dimensional with a large number of parameters". Please see at P3, L59-60.

**8. Comment**: *Lines 62-63: 'These quantitative criteria' refer to what?*

   **Response:** We have reorganized this paragraph, and deleted the original statement. Please see at P3, L67-68.

**9. Comment**: *Line 70: performances.*

   **Response:** Thanks. We have reorganized this paragraph and corrected the statement. Please see at P3, L75.

**10. Comment**: *Line 71: remove 'the' from 'the SW model'*

   **Response:** We have removed 'the' from 'the SW model'.

**11. Comment**: *Lines 71-72: please rephase this sentence.*

   **Response:** We have changed the sentence "Ershadi et al. (2014) evaluated the surface energy balance system (SEBS), PM, PT-JPL (a modified Priestley-Taylor model), and AA models." This can be seen at P3, L76-77.

**12. Comment**: *Line 73: should be model ranking? Please check the terminology.*

   **Response:** Thanks. We have corrected this mistake at P3, L69.

**13. Comment**: *Lines 75-76: unclear, significant variability of model performances?*

   **Response:** Considering the comments, we have changed "significant" to "considerable". This can be seen at P3, L80.

**14. Comment**: *Lines 92-93: been?*

   **Response:** We have corrected the sentence.

**15. Comment**: *Lines 102-103: add a reference*

   **Response:** We have added the reference "(Vrugt et al., 2008, 2009)" at P5, L119-120.

   **Once again, thank you very much for your comments and suggestions.**

**Bayesian performance evaluation of evapotranspiration models: a case study based on eddy covariance system of a maize field in northwestern China**

Guoxiao Wei [1, 2] [*], Xiaoying Zhang [3, *], Ming Ye [4], Ning Yue [1,2], Fei Kan [1,2]

[1] Key Laboratory of Western China's Environmental System (Ministry of Education), Lanzhou University, China, 730000
[2] School of Earth and Environmental Sciences, Lanzhou University, China, 730000
[3] Construct Engineering College, Jilin University, China, 130400
[4] Department of Earth, Ocean, and Atmospheric Science, Florida State University, USA, 32306
* Corresponding author: xiaoyingzh@jlu.edu.cn.

**Abstract**

Evapotranspiration (ET) is a major component of the land surface process involved in energy fluxes and energy balance, especially in the hydrological cycle of agricultural ecosystems. While many models have been developed as powerful tools to estimate ET, there is no agreement on which model best describing the loss of water to the atmosphere. In this study, we present a solid study to evaluate four widely used ET models and their parameter contributions  using half-hourly ET observations obtained at a spring maize field in an arid region. The four tested models are  Shuttleworth Wallace (SW) model, Penman-Monteith (PM) model, Priestley-Taylor and Flint-Childs (PT-FC) model, and Advection-Aridity (AA) model. The parameters in each model were first calibrated using DiffeRential Evolution Adaptive Metropolis (DREAM) algorithm, and then were analyzed to identify their impacts on the model performance. The Bayesian model evidence (BME) approach, was further adopted to select the optimal model by incorporating the mathematically rigorous thermodynamic integration algorithm.~~The BME-based model ranking (from the best to the worst) is SW, PM, PT-FC, and AA. The residuals between observations and corresponding model simulations are also analyzed, and the same model ranking is also obtained by using residual-based statistics, i.e., the coefficient of determination ($R^2$), index of agreement (IA), root mean square error (RMSE) and model efficiency (EF). The PM and SW models overestimate ET, whereas the PT-FC and AA models underestimate ET in the study period. The four models also underestimate ET during the periods of partial crop cover. Especially during the late maturity stage, the PT-FC and AA models consistently produce an underestimation, and provide the worst simulated ET. As a result, at the half-hourly time scale, the SW model is the best model and recommend as the first choice for evaluating ET of spring maize in arid desert oasis areas.icient is well constrained in the SW model and poorly constrained in the PM model, but not considered in PT-FC and AA models.~~ This is the main reason that the SW model outperforming the other models. Although the SW model with seven parameters is sophisticated, it's good fitting to observations can counterbalance its higher complexity. In addition, the discrepancies between observations and model simulations were evaluated using traditional error metrics. The mismatch analysis indicated 
[revised manuscript text omitted]
_{\mathrm{a}} \\ \dfrac{\theta-\theta_{\mathrm{b}}}{\theta_{\mathrm{a}}-\theta_{\mathrm{b}}} & \theta_{\mathrm{b}} < \theta < \theta_{\mathrm{a}} \\ 0 & \theta < \theta_{\mathrm{b}} \end{cases} \tag{\cancel{6}5}$$

and $\theta_a$  is set as $\theta_a = 0.75\ \theta_b$. Aerodynamic conductance $g_a$ is calculated as:

$$g_{\mathrm{a}} = \frac{k^2 u_{\mathrm{m}}}{\ln\left[(z_{\mathrm{m}}-d)/z_{0\mathrm{m}}\right]\ln\left[(z_{\mathrm{m}}-d)/z_{0\mathrm{v}}\right]} \tag{\cancel{7}6}$$

where the quantities $d$, $z_{0m}$ and $z_{0v}$ are calculated using $d = 2h/3$, $z_{0m} = 0.123h$ and $z_{0v} = 0.1z_{0m}$ (Allen 1998).

**2.3.2. Shuttleworth-Wallace (SW) model**

The SW model comprises a one-dimensional model of plant transpiration and a one-dimensional model of
soil evaporation. The two terms are calculated by the following equations:

$$\lambda\mathrm{ET} = \lambda E + \lambda T = C_{\mathrm{s}}\mathrm{ET}_{\mathrm{s}} + C_{\mathrm{c}}\mathrm{ET}_{\mathrm{c}} \tag{7}$$

$$\mathrm{ET}_{\mathrm{s}} = \frac{\Delta A + \left\{\rho C_{\mathrm{p}}(e_{\mathrm{s}}-e_{\mathrm{a}})-\Delta r_{\mathrm{a}}^{\mathrm{s}}\left(A-A_{\mathrm{s}}\right)\right\}/\left(r_{\mathrm{a}}^{\mathrm{a}}+r_{\mathrm{a}}^{\mathrm{s}}\right)}{\Delta + \gamma\left\{1+r_{\mathrm{s}}^{\mathrm{s}}/\left(r_{\mathrm{a}}^{\mathrm{a}}+r_{\mathrm{a}}^{\mathrm{s}}\right)\right\}} \tag{8}$$

$$\mathrm{ET}_{\mathrm{c}} = \frac{\Delta A + \left\{\rho C_{\mathrm{p}}(e_{\mathrm{s}}-e_{\mathrm{a}})-\Delta r^{\mathrm{c}}A_{\mathrm{s}}\right\}/\left(r_{\mathrm{a}}^{\mathrm{a}}+r_{\mathrm{a}}^{\mathrm{c}}\right)}{\Delta + \gamma\left\{1+r_{\mathrm{s}}^{\mathrm{c}}/\left(r_{\mathrm{a}}^{\mathrm{a}}+r_{\mathrm{a}}^{\mathrm{c}}\right)\right\}} \tag{9}$$

where the available energy input above the soil surface is defined as $A_{\mathrm{s}} = R_{\mathrm{ns}} - G$.

$R_{ns}$ can be calculated using the Beer's law relationship:

$$R_{\mathrm{ns}} = R_{\mathrm{n}}\exp\left(-K_{\mathrm{a}}\mathrm{LAI}\right) \tag{10}$$

The coefficients $C_s$ and $C_c$ are obtained as follows:

$$C_{\mathrm{s}} = \left\{ 1 + R_{\mathrm{s}} R_{\mathrm{a}} \big/ R_{\mathrm{c}} \left( R_{\mathrm{s}} + R_{\mathrm{a}} \right) \right\}^{-1} \tag{11}$$

$$C_{\mathrm{c}} = \left\{ 1 + R_{\mathrm{c}} R_{\mathrm{a}} \big/ R_{\mathrm{s}} \left( R_{\mathrm{c}} + R_{\mathrm{a}} \right) \right\}^{-1} \tag{12}$$

where

$$R_{\mathrm{a}} = \left( \Delta + \gamma \right) r_{\mathrm{a}}^{\mathrm{a}} \tag{13}$$

$$R_{\mathrm{s}} = \left( \Delta + \gamma \right) r_{\mathrm{a}}^{\mathrm{s}} + \gamma r_{\mathrm{s}}^{\mathrm{s}} \tag{14}$$

$$R_{\mathrm{c}} = \left( \Delta + \gamma \right) r_{\mathrm{a}}^{\mathrm{c}} + \gamma r_{\mathrm{s}}^{\mathrm{c}} \tag{15}$$

Soil surface resistance is expressed as:

$$r_{\mathrm{s}}^{\mathrm{s}} = \mathrm{exp} \left( b_{1} - b_{2} \frac{\theta}{\theta_{\mathrm{s}}} \right) \tag{16}$$

[revised manuscript text omitted]

---

## Author Response (AR2)

**Manuscript Number:** hess-2018-430

**Title**: Bayesian performance evaluation of evapotranspiration models based on eddy covariance systems in the arid region

**Corresponding Author:** Xiaoying Zhang

**Authors:** GuoxiaoWei, Xiaoying Zhang, Ming Ye , Ning Yue, Fei Kan

**Dear Editor,**

On behalf of my co-authors, we thank you very much for giving us a positive comment on our revised manuscript. We are grateful to the editors and reviewers for their constructive suggestions on our manuscript (hess-2018-430).

Based on the reviewer's comment, we have briefed our manuscript title to "Bayesian performance evaluation of evapotranspiration models based on eddy covariance systems in the arid region". Meanwhile, we have corrected the technical issues through the manuscript. Attached please find the revised version, which we would like to submit for your kind consideration.

We would like to express our great appreciation to you and reviewers for comments on our paper. Looking forward to hearing from you.

Thank you and best regards.

Yours sincerely,

Xiaoying Zhang
Corresponding author:
Name Xiaoying Zhang
E-mail: xiaoyingzh@jlu.edu.cn

**Responds to the comments:**

**Referee #1:**

**General comments:**

**1. Comment**: *The title of this manuscript should be more general.*

Response: Thanks for the comment, we have modified the manuscript title to "Bayesian performance evaluation of evapotranspiration models based on eddy covariance systems in the arid region".

**Bayesian performance evaluation of evapotranspiration models: a case study based on**

**eddy covariance systems in the arid region of a maize field in northwestern China**

Guoxiao Wei [1, 2], Xiaoying Zhang [3, *], Ming Ye [4], Ning Yue [1,2], Fei Kan [1,2]

[1] Key Laboratory of Western China's Environmental System (Ministry of Education), Lanzhou University,
China, 730000
[2] School of Earth and Environmental Sciences, Lanzhou University, China, 730000
[3] Construct Engineering College, Jilin University, China, 130400
[4] Department of Earth, Ocean, and Atmospheric Science, Florida State University, USA, 32306
* Corresponding author: xiaoyingzh@jlu.edu.cn.

**Abstract**

Evapotranspiration (ET) is a major component of the land surface process involved in energy fluxes and energy balance, especially in the hydrological cycle of agricultural ecosystems. While many models have been developed as powerful tools to simulateestimate ET, there is no agreement on which model best describing the loss of water to the atmosphere. This study focuses on two aspects, evaluating the performance of four widely used ET models, and identifying parameters, as well the physical mechanisms that have significant impacts on the model performance. The four tested models are Shuttleworth Wallace (SW) model, Penman-Monteith (PM)

model, Priestley-Taylor and Flint-Childs (PT-FC) model, and Advection-Aridity (AA) model. The parameters in each model were first calibrated using DiffeRential Evolution Adaptive Metropolis (DREAM) algorithm, and then were analyzed to identify their impacts on the model performance. 
[revised manuscript text omitted]

$$\text{ET}_s = \frac{\Delta A + \left\{\rho C_p (e_s - e_a) - \Delta r_a^s (A - A_s)\right\} / (r_a^a + r_a^s)}{\Delta + \gamma \left\{1 + r_s^s / (r_a^a + r_a^s)\right\}} \tag{8}$$

$$\text{ET}_c = \frac{\Delta A + \left\{\rho C_p (e_s - e_a) - \Delta r_a^c A_s\right\} / (r_a^a + r_a^c)}{\Delta + \gamma \left\{1 + r_s^c / (r_a^a + r_a^c)\right\}} \tag{9}$$

where the available energy input above the soil surface is defined as $A_s = R_{ns} - G$.

$R_{ns}$ can be calculated using the Beer's law relationship:

$$R_{ns} = R_n \exp\left(-K_a \text{LAI}\right) \tag{10}$$

The coefficients $C_s$ and $C_c$ are obtained as follows:

$$C_s = \left\{1 + R_s R_a / R_c (R_s + R_a)\right\}^{-1} \tag{11}$$

$$C_c = \left\{1 + R_c R_a / R_s (R_c + R_a)\right\}^{-1} \tag{12}$$

where

$$R_a = (\Delta + \gamma) r_a^a \tag{13}$$

$$R_s = (\Delta + \gamma) r_a^s + \gamma r_s^s \tag{14}$$

$$R_c = (\Delta + \gamma) r_a^c + \gamma r_s^c \tag{15}$$

[revised manuscript text omitted]